# Random rotational embedding Bayesian optimization for human-in-the-loop personalized music generation

Miguel Marcos *, Lorenzo Mur-Labadia, Ruben Martinez-Cantin

Departamento de Informática e Ingeniería de Sistemas, Instituto de Investigación en Ingeniería de Aragón (I3A), Universidad de Zaragoza, Zaragoza, Spain

* m.marcos@unizar.es

## Abstract

Generative deep learning models, such as those used for music generation, can produce a wide variety of results based on perturbations of random points in their latent space. User preferences can be incorporated in the generative process by replacing this random sampling with a personalized query. Bayesian optimization, a sample-efficient nonlinear optimization method, is the gold standard for human-in-the-loop optimization problems, such as finding this query. In this paper, we present *random rotational embedding Bayesian optimization* (ROMBO). This novel method can efficiently sample and optimize high-dimensional spaces with rotational symmetries, like the Gaussian latent spaces found in generative models. ROMBO works by embedding a low-dimensional Gaussian search space into a high-dimensional one through random rotations. Our method outperforms several baselines, including other high-dimensional Bayesian optimization variants. We evaluate our algorithm through a music generation task. Our evaluation includes both simulated experiments and real user feedback. Our results show that ROMBO can perform efficient personalization of a generative deep learning model. The main contributions of our paper are: we introduce a novel embedding strategy for Bayesian optimization in high-dimensional Gaussian sample spaces; achieve a consistently better performance throughout optimization with respect to baselines, with a final loss reduction of 16%-31% in simulation; and complement our simulated evaluations with a study with human volunteers (n = 16). Users working with our music generation pipeline find new favorite pieces 40% more often, 16% faster, and spend 18% less time on pieces they dislike than when randomly querying the model. These results, along with a final survey, demonstrate great performance and satisfaction, even among users with particular tastes.

**Data availability statement:** All data and code developed for this project, evaluation of the method and recreation of results are available GitHub public repository under the URL 'https://github.com/mikceroese/ROMBO'. The site includes an online MIDI player where generated content may be played.

**Funding:** This research was financially supported by Spanish government (www.aei.gob.es) projects funding awards (PID2021-125209OB-I00 and PID2024-158322OB-I00) received by MM, LM, and RM. This study was also financially supported by the Aragon government (www.aragon.es) in the form of a funding award (DGA T45_23R) received by MM, LM, and RM. The funders had no role in study design, data collection and analysis, decision to publish, or preparation of the manuscript.

**Competing interests:** The authors have declared that no competing interests exist.

## Introduction

Artificial intelligence systems for music generation can be useful tools in a wide range of applications. For example, initiating amateur musicians [1–5], helping in music therapy for medical treatments [6,7], and even automatically remixing or rearranging songs. The latest music-generative models [1,2] can produce several minutes of music from a text prompt. Generative models for other modalities can also produce realistic images, text, etc. However, most models [8,9] require complex *prompt engineering* to modify the results to get the desired outcome. Even with a good prompt, generative models use a random sample of a *latent space* to provide variability in the outcome, resulting in unexpected results. For music, obtaining a specific output with the desired rhythm, tension, or emotion flow can be challenging [10].

We propose to formulate the generative process as an optimization problem, where the objective is to find the desired outcome based on simple ratings, instead of complex *prompts*. Instead of drawing random samples, we optimize the latent space sample such that it maximizes the user's preference. Importantly, we seek to find a maximum in a limited number of trials to minimize user interaction. Besides, we want our optimization approach to be *agnostic* to the generative model used, treating the model and the user feedback as a *black-box function*.

**Bayesian optimization** (BO) is the state-of-the-art approach for data-efficient black-box global optimization [11–13] and, therefore, it excels in human-feedback in-the-loop optimization [14–18]. BO builds a surrogate model iteratively, based on observations. This model acts as a memory of previous queries, quickly reducing the search space as we gather data. Meanwhile, an acquisition function over this surrogate model locates potential optima, focusing the search around promising candidates. Using the model's information, combined with optimal decision theory, results in a trade-off between exploration and exploitation of the search space. BO is also robust to observation noise, which is a typical limitation of human feedback. By using a probabilistic surrogate model, we can encode and take into account noise in the query responses to provide a robust solution. This is true even in the presence of spurious data [19] or mismodeling errors [20].

In general, any global optimization algorithm, including BO, is limited to low-dimensional problems and fails to address problems with hundreds of dimensions in the search space. **High-dimensional problems** require an exponentially large number of observations or a strong prior over the problem structure [11]. For example, if the effective dimensionality of the problem is within a low-dimensional manifold [21,22], the problem can be decomposed into additive subspaces [23,24], or we can address the problem with local models [25].

In this work, we want to optimize a function over the latent space of a generative model, which is a high-dimensional Gaussian space. This latent space does not fall in any of the structures assumed in high-dimensional BO methods, such as linear or box-bounded constraints [21]. We present a **novel BO algorithm** adapted to high-dimensional Gaussian sampling spaces, called *random rotational embedding Bayesian optimization* (ROMBO). Our algorithm uses random rotation matrices to map the high-dimensional problem to a lower-dimensional space where BO can be

used effectively. By using random rotation matrices, the Gaussian structure of the sampling space is preserved in the lower-dimensional space. We present both theoretical and numerical results that show the advantages of our algorithm over previous algorithms.

Using a music generative model and our algorithm, we build a **human-in-the-loop music generation pipeline**, featured in Fig 1. We use a lightweight and efficient deep learning model that runs on inexpensive hardware [26]. The model generates multi-instrument music compositions (MIDI files) and it is trained to provide realistic rock compositions featuring several instruments and harmonic tracks, with a diverse variety of results. Users rate the generated compositions. The ratings are then used by ROMBO to explore the sampling space to find the optimum using few samples –i.e., few human interactions–.

By reducing the dimensionality of the model's latent space, we can efficiently explore it or further refine the preferred sample for the user. Moreover, since BO and ROMBO are model-agnostic, our proposal works for any generative model with a Gaussian latent space.

This **human-feedback for generative models** strategy is related to how modern large-language models (LLMs) complement their pre-trained architectures with reinforcement learning with human feedback (RLHF) [27,28]. In these models, the large generative model is fine-tuned to optimize the reward estimated from previous interactions with humans to address their common preferences. In contrast, our approach personalizes the model output interactively and independently for each individual, focusing on data efficiency and privacy. Therefore, if an individual repeated the process with different preferences –e.g., in a different mood– the results would change accordingly. This work is, to our knowledge, the second to apply BO to music generation with human feedback [29], and the first one to ever try to optimize high-complexity harmonic compositions for several instruments.

Our **hypothesis** is twofold. First, Bayesian optimization using rotational embeddings outperforms the state-of-the-art when faced with a high-dimensional Gaussian input space, such as the sampling space in generative models. Second, Bayesian optimization with our rotational embedding can be used to personalize preferences in generative models.

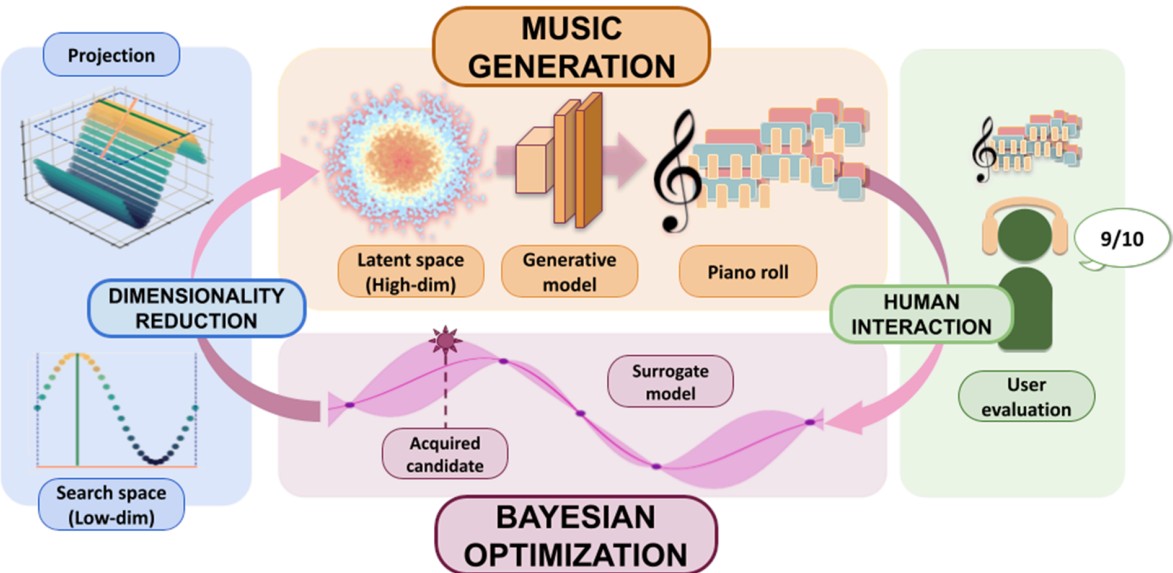

**Fig 1**. **Overview of our proposed pipeline.** We sample a low-dimensional search space, which is remapped onto a music generation model's high-dimensional latent space. After playing the resultant piece, we use a user-given grade to build a surrogate model and efficiently search for new candidate points using Bayesian Optimization, adapting to a user's taste in real time without retraining the model.

The first hypothesis is validated both mathematically and numerically. For the numerical validation, we test ROMBO against a collection of baselines. User preferences are simulated as randomly selected target songs that the optimizer tries to match as closely as possible. We show how our method, specifically designed for high-dimensional Gaussian domains, outperforms all the baselines.

Our second hypothesis is tested in an experiment with actual human feedback, featuring 16 participants. A group using our music generation pipeline personalized with ROMBO is compared to another using the generative model queried without personalization. People in the first group refine or find new favorite songs more often, and navigate through uninteresting queries for less time. We also explore the features of the final queries and user responses in a final Likert survey.

In summary, our main contributions are:

- A novel approach for performing Bayesian optimization in high-dimensional Gaussian search spaces, especially useful for generative models, outperforming other Bayesian optimization baselines and dimensionality reduction techniques.
- A sample-efficient, personalized music generation method based on subjective evaluations, implemented using Bayesian optimization and our own version of a gold-standard music generation model.
- A collection of human-driven experiments proving both the need for model personalization and the effectiveness of our method.

## Related works

### Generative models

The goal of a generative model is to capture the distribution of a set of data to then create new samples in said distribution. Within this category, we find Generative Adversarial Networks [30] (GANs); their later successors, Wasserstein GANs [31] (WGANs); and Variational Autoencoders [32,33] (VAEs). All these architectures use two networks that either compete or collaborate, ultimately creating a latent space to sample from later. The last two of them are employed in our method. Among later architectures, we encounter diffusion models [9], which learn to generate data from noise by learning a denoising function. They condition the process on a target class, descriptor, or prompt to obtain their results. Though these models are widely used for image generation, other data types, like text or music, can be generated with them. In fact, there are specialized architectures for different data types. For example, Generative Pretrained Transformers (GPTs [34,35]) exploit attention mechanisms [8] to generate text word by word, conditioning it on both a prompt and their previously generated text. They are often tuned with Reinforcement Learning from Human Feedback [27,28,36] (RLHF), which profiles preferred outputs based on human judgment, in a way similar to our proposed method.

For the particular case of artificial music composition and generation (referred to as just "music generation", onward), the first artificial neural networks used for composition started with MultiLayered Perceptrons (MLPs) [37,38], then evolved to Short-Term Memory Networks [39,40] (LSTMs) and into GANs [26,41]. Found among the latter, MuseGAN [26] is a WGAN trained over MIDI files to create piano rolls, binary matrices with positive values representing each sounding note, with up to five different instruments. Coconet [42], a later model that also works on composition, approaches the process by trying to imitate human composers. Instead of writing in the direction of time like other models, a Neural Autoregressive Density Estimator (NADE) [43] fills gaps in Bach chorales. This teaches the model to rewrite and iterate over its pieces, not tying the process to any causal direction. Later models shift focus from the symbolic domain to the audio domain, abstracting the composition process. MelNet [44], a model contemporary to Coconet, generates realistic piano solos and is trained with Mel-spectrograms, 2D decompositions of the soundwave on different frequencies remapped to a quasi-logarithmical perception-based scale. Later in time, we encounter models that exploit the Transformer architecture, like AudioCraft, which combines audio compression [3] and text-to-sound generation [4,5], or Jukebox [1], which can also generate vocal parts from written lyrics. One of the latest trending examples is Bark [2], which is the text-to-speech model used in the AI music generator Suno and is capable of generating music and sound effects next to the lyrics from the text

prompt. Lastly, generative models should achieve results that evoke certain emotions, especially if asked to generate personalized content. Hence, architectures that focus on recognising these emotions, such as the graph neural networks HGLER [45] or ENMI [46], can become important additions to music generation applications.

## Bayesian optimization

Bayesian optimization has been widely used in many applications and scenarios. Examples include hyperparameter tuning for neural networks [47], robust policy search, planning, and motion in robotics [20,48,49], autonomous systems [12], game agent optimization [50], aircraft design [51], physics modeling and simulation [52], biochemistry [53], ecology [54], and medicine [55–57] among others.

Some works focus on Bayesian optimization as a way to take into account human judgment for tasks like image generation [58], photography color enhancement [16,59], material appearance modeling [16,60], procedural animations [17] and prosthesis calibration [15,61]. These cases usually lean towards preferential Bayesian optimization (PBO), which captures relations between evaluations instead of assigning scalars individually. This takes care of some biases, such as drift or anchoring, when working with human judgment. PBO has been used for the optimization of music generation [29], where users manipulate an interface for melody composition, onto which they can select a two-bar section for the system to propose variations, which they can also edit themselves. However, high-dimensional PBO is still an open problem with few preliminary results addressing it [62] limited to 1D preferences.

High-dimensional vanilla BO, on the other hand, is a well-known issue, and many techniques tackle its limitations [63]. Some assume the effective dimensionality of the problem is smaller than its domain, and hence some dimensions may be uninformative. These dimensions may be axis-aligned [64,65] or not [21,22]. Thus, some authors opt to cull down the search space [21], while others enforce sparse priors on the dimensions [65,66]. Another option is breaking down the GP into a sum of several smaller and independent component functions, which receives the name of Additive kernels [24,67, 68]. This approach makes the total complexity of fitting all kernels lower than fitting a single global kernel. Some others choose to change the exploration strategy. Some works propose to impose locality on the optimization [25], which simplifies the process, while others prefer transforming the geometry of the search space by making it non-Euclidean, e.g., through cylindrical kernels [69], which effectively expands the center of the search space while shrinking the borders.

## Methods

We present a pipeline for personalized music generation leveraging human feedback. Our approach, portrayed in Fig 1, has four essential pieces: A deep-learning model for music generation, Bayesian optimization as our sample-efficient query generation for finding a high-quality personalized composition, a dimensionality reduction process to make the optimization tractable, and a user interface to evaluate the queries (compositions) based on user taste. Below, we explain the details of each of these main parts.

## Music generation model

Our method's foundation is a music generation model. Following former literature [29] on music-oriented BO, we chose symbolic music generation as the domain of our problem.

Symbolic music is most often represented in score sheets: Several instruments play different parts of the score, each with their own dynamics, but interacting with each other. Notes unfold along time following structural patterns, grouping in bars and forming phrases. Lastly, notes are played in different pitches, and stack forming arpeggios and chords.

Our data is a set of piano rolls, which imitate the structure of score sheets. A piano roll is a multi-dimensional binary tensor, which is made of $I$ instrument tracks that play for $B$ bars of $T$ timesteps. Instruments play within a range of $P$ pitches, with zeros meaning silence, and ones meaning sound. These are obtained from and convertible to MIDI files. Mathematically, a piano roll is defined as $\Psi \in [0,1]^{I \times B \times T \times P}$. This multi-dimensional setup allows for several instruments to

play several notes at the same time. Compared to melodies, where only up to one note is playing at a time, it increases the range of possible generations, as well as complexity.

For our selected model, we take inspiration from MuseGAN [26], a WGAN capable of generating piano rolls for up to five instruments. Formally, a WGAN is composed of two networks, a generator $G$, and a critic $C$, facing each other in a minmax game

$$\min_G \max_C \mathbb{E}_{x \sim P_X}[C(x)] - \mathbb{E}_{z \sim P_Z}[C(G(z))] \tag{1}$$

where $G$ takes an input $z$ from a latent space $Z$ that follows a distribution $P_Z$, and creates counterfeit samples $G(z)$ that should emulate the real samples $x$ from the real distribution $P_X$. Meanwhile, $C$ examines both the real and counterfeit samples and outputs values $C(x)$ and $C(G(z))$ for each of them. $C$ must maximize the difference between the scores of the real samples and the fakes, while $G$ must reduce it. To accelerate convergence, a gradient penalty [70] term is added to the critic's objective function, which becomes

$$\mathbb{E}_{x \sim P_X}[C(x)] - \mathbb{E}_{z \sim P_Z}[C(G(z))] + \mathbb{E}_{g \sim P_G}[(\nabla_g \|g\| - 1)^2] \tag{2}$$

where $P_G$ is defined uniformly sampling interpolations between sample pairs $(x, G(z))$.

Our model, which we named PowGAN, is a lighter version of the "Composer" model of the original MuseGAN that can generate piano rolls for drums, guitar, and bass (called a 'Power trio' in rock music, hence the name), taking Gaussian noise as input. The original MuseGAN model also included strings and piano, which we removed since we deemed five instruments were difficult for humans (potentially non-experts) to evaluate within sensible time limits. The piano rolls used for training the model are a subset of the Lakh Pianoroll Dataset [26], which is derived from the Lakh MIDI Dataset [71]. The reasoning for which tracks to remove comes from analyzing this dataset. The piano track was the one with the most silent segments, and the strings track included all other instruments other than the five listed.

Regarding the architecture, both the generator and critic are multi-stream 3D Convolutional Neural Networks. A schematic overview of the generator is depicted in Fig 2. The generator's architecture consists of an initial part shared among instruments that upsamples the input and later splits into private streams for each instrument. Each instrument stream performs inverse-3D convolutions over bars, time, and pitch, and is also split into two halves where dimension expansion occurs in different orders: time-first, and pitch-first. Each stream's halves are merged at the end to form the final track for each instrument. The critic follows an architecture mostly symmetric to the generator, with the addition of two global streams that capture harmony and rhythm features for all the composition [26].

The features of the generated pieces, and the adequacy of the distribution and the model, are profiled through the following metrics:

- Empty Bar Ratio (EBR). Measures the percentage of completely silent bars. It ranges from 0 to 1. Applies to all tracks.
- Used Pitch Classes (UPC). Measures the number of different notes used in each pitched track. Classes circle back every scale, so the same note played in different scales counts as only one note. It ranges from 0 to 12. Applies to Guitar and Bass.
- Tonal Distance [72] (TD). Measures harmonic relations between pitched tracks. Intuitively, the lower the TD, the stronger the relation and the better the tracks sound together. Applies to the Guitar-Bass pair.

### Bayesian optimization

Bayesian optimization is our tool of choice for exploring the latent space of the developed model. BO seeks to optimize an unknown function $f$ using a probabilistic surrogate model $P(f)$, commonly a Gaussian process (GP), and an acquisition function.

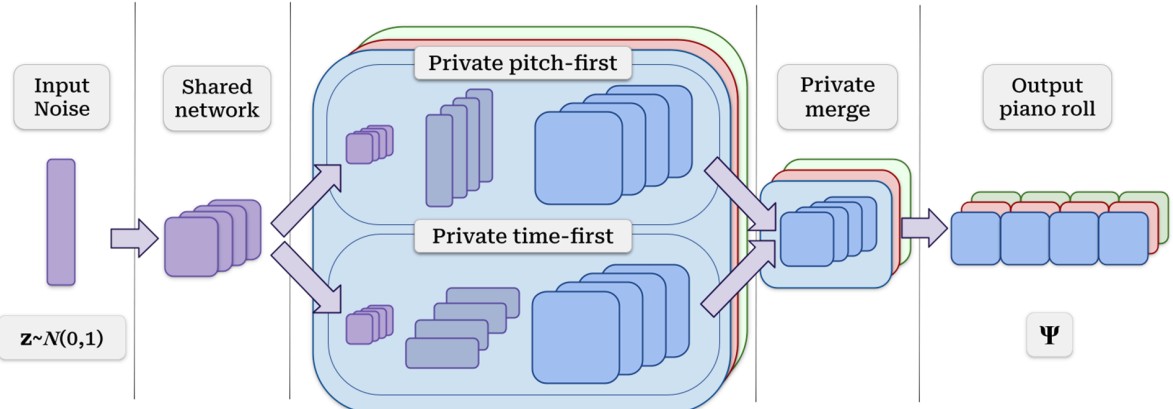

**Fig 2. PowGAN's generator architecture overview.** The model takes a Gaussian noise vector, upsamples it through the shared network, and creates common features that are replicated for all instruments. Features run through private parallel instrument streams. Each instrument stream (drums, guitar, and bass) splits in two, upsamples these features, and merges again, each creating the bars for the corresponding track. Bars in each track are concatenated and form the final piano roll $\Psi$.

A Gaussian process is a collection of random variables such that every finite subset of them has a multivariate normal distribution, and it is defined by its mean $m(\cdot)$ and covariance $k(\cdot, \cdot)$ functions. The GP is built on a dataset $\mathcal{D}_n$ of $n$ previous observations $(\mathbf{x}_i, y_i)$ of $f$, and its posterior probability $P(f|D)$ is updated every time a new observation is acquired. The posterior model allows predictions $y_q$ at query points $\mathbf{x}_q$ which follow a normal distribution $y_q \sim \mathcal{N}(\mu(\mathbf{x}_q), \sigma^2(\mathbf{x}_q))$ such that

$$\mu(\mathbf{x}_q) = \mathbf{k}^T(\mathbf{K} + \mathbf{I}\sigma_n^2)^{-1}\mathbf{y}$$
$$\sigma^2(\mathbf{x}_q) = k(\mathbf{x}_q, \mathbf{x}_q) - \mathbf{k}^T(\mathbf{K} + \mathbf{I}\sigma_n^2)^{-1}\mathbf{k} \tag{3}$$

where $\mathbf{y}$ is the vector of evaluation results, $\mathbf{k}$ is a vector with components $k_i = k(\mathbf{x}_q, \mathbf{x}_i)$, $\mathbf{K}$ is the kernel matrix with components $K_{ij} = k(\mathbf{x}_i, \mathbf{x}_j)$ for $i, j \in \{1, ..., n\}$ and $\sigma_n^2$ is the noise variance. There exist multiple options for $k(\cdot, \cdot)$, but in this work we use the Matèrn kernel $\nu = \frac{5}{2}$ which is known to perform well for Bayesian optimization [12,47]. The Matèrn kernel $\nu = \frac{5}{2}$ is defined as:

$$k_{5/2}(\mathbf{x}, \mathbf{x}'|\theta) = \sigma_s^2 \left(1 + \sqrt{5}r + \frac{5}{3}r^2\right)\exp\left(-\sqrt{5}r\right) \tag{4}$$

where $\sigma_s^2$ is the signal variance and $r$ is the Mahalanobis distance between points $\mathbf{x}$ and $\mathbf{x}'$. In particular, we use the Matèrn kernel with *automatic relevance determination* (ARD), such that:

$$r = \sqrt{(\mathbf{x} - \mathbf{x}')^T\Lambda(\mathbf{x} - \mathbf{x}')} \qquad \text{with} \qquad \Lambda = \text{diag}(\lambda^{-1}) \tag{5}$$

where $\lambda$ is a vector of the lengthcales of the kernel resulting in a diagonal matrix $\Lambda$ with one lengthscale per dimension of $\mathbf{x}$. Each lenghtscale measures the scale of the distance in that dimension. For equally distant coordinates $\mathbf{x}_1$ and $\mathbf{x}'_1$, a larger lengthscale results in a higher Mahalanobis distance and a higher correlation between $\mathbf{x}_1$ and $\mathbf{x}'_1$. For regression and optimization, this high correlation means that knowing $\mathbf{x}_1$ already allows you to predict $\mathbf{x}'_1$ with high accuracy and *vice versa*. Therefore, sampling that coordinate becomes unnecessary, resulting in irrelevant dimensions. Contrary, a small lengthscale means a lower correlation, making $\mathbf{x}_1$ independent of $\mathbf{x}'_1$. In practice, this requires having multiple samples of that dimension to correctly approximate the function, resulting in a relevant dimension for regression and optimization.

The kernel hyperparameters, which includes the lengthscales, signal variance and noise variance $\theta = [\sigma_n^2, \sigma_s^2, \lambda]$ are usually estimated through an *empirical Bayesian* approach, meaning they are estimated from either *maximum marginal likelihood* (ML) or *maximum a posteriori*: $\theta^* = \arg\max_\theta \sum_i \log p(y_i|\mathbf{x}_i, \theta)$. However, empirical Bayes results in an overconfident uncertainty estimation of the Gaussian process as it ignores the uncertainty of the lenghtscale estimates [12,47]. Instead, we estimate the lengthscale posterior distribution through a *fully Bayesian* approach based on Markov chain Monte Carlo (MCMC) that with better uncertainty estimation and more robust results [12,47,73]. While MCMC is more computationally intensive than ML, the extra cost is negligible compared to the cost of evaluating $f(\mathbf{x})$. Note that in this work, $f(\mathbf{x})$ corresponds to generating a piece of music, reproduce it to the user and wait for the rating.

In addition to the surrogate model, BO also uses an acquisition function $a(\mathbf{x}, P(f))$ for efficiently choosing the next evaluation point $\mathbf{x}_{t+1}$ at each step $t$ of the algorithm, such that $\mathbf{x}_{t+1} = \arg\max_\mathbf{x} a(\mathbf{x}, P(f|\mathcal{D}_t))$. Our choice for $a(\mathbf{x}, P(f))$ is the Expected Improvement (EI) function

$$\text{EI}(\mathbf{x_q}) = \mathbb{E}_{p(y_q|\mathbf{x_q}, \theta)}[\text{I}(\mathbf{x_q})] = (\rho - \mu(x_q))\Phi(z) + \sigma(x_q)\phi(z)$$
$$z = (\rho - \mu(x_q))/\sigma(x_q)$$

(6)

where $\text{I}(\mathbf{x_q}) = \max(0, \rho - f(\mathbf{x_q}))$ is the improvement function, $\rho$ is the current best result among the observations of the GP, $\phi$ and $\Phi$ are the Gaussian probability density function (PDF) and cumulative density function (CDF), and $\mu(x)$ and $\sigma(x)$ are obtained from Eq 3.

A representative initial dataset is key to fitting a meaningful initial model. We obtain our initial query points through Sobol sampling [74], a quasi-random series that ensures a good cover of the domain. After the initial evaluations, the algorithm runs iteratively by fitting the GP to all available observations and selecting new query points at the maxima of the acquisition function, as portrayed in Fig 3, adding the observations to the dataset. A pseudocode overview is provided in Algorithm 1.

**Gaussian spaces.** In order to explore a Gaussian space through BO, we need to adapt it from its assumed box-bounded search space into a Gaussian one. Some works [75] extend BO to iteratively increase the bounded volume, potentially to infinity, though it is still a hypercube. In our Gaussian latent space, the distribution of the data is concentrated around the origin, which means our search should be more thorough around this center area, as we can expect greater variation.

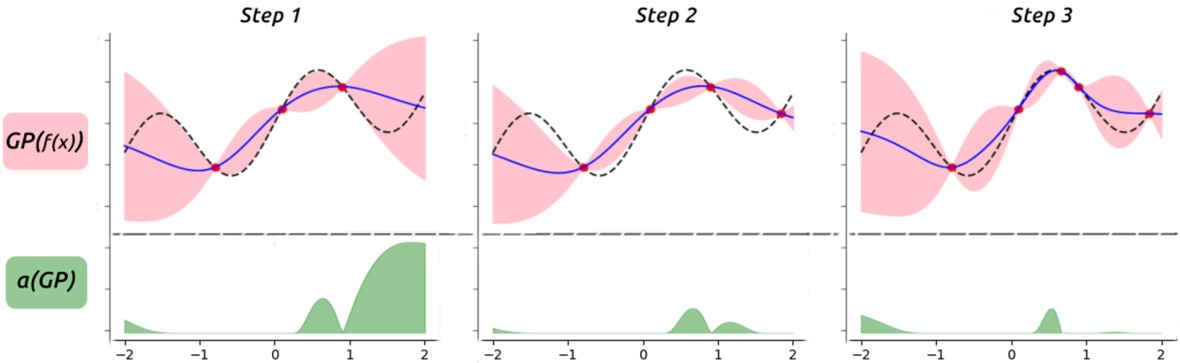

**Fig 3. Bayesian optimization overview.** The GP (mean in blue, variance in pink) is fit to all previous observations (red dots) that follow the target function (dashed line) and the acquisition function $a$ (EI, in green) indicates where the next query will be sampled.

### Algorithm 1 Bayesian optimization.

```
Require: Objective function f, acquisition function a, budget N
 1: Initialize data set D = (X, y) with the data points X = {x₁, x₂, ..., xₗ} and their responses y = {y₁, y₂, ..., yₗ},
    using a preliminary design.
 2: for t = 1, 2, ..., N do
 3:     Fit the surrogate model to D: P(f|X, y) ∝ P(X, y|f)P(f)
 4:     xₜ = arg maxₓ C(x|P(f|X, y))
 5:     yₜ = f(xₜ) + σₙ  // With σₙ observation noise
 6:     D = D ∪ (xₜ, yₜ)
 7: end for
 8: Return arg maxₓ(y|(x, y) ∈ D)
```

To solve this, we use the Box-Muller (BM) transform to map an exploration space $[0,1]^n$ to a Gaussian one $\mathcal{N}(\mathbf{0}, \mathbf{I}^n)$. We show an example of how this transformation affects a two-dimensional space in Fig 4, though it works for any even $n$ number of dimensions.

## High-dimensional Bayesian optimization

PowGAN's 256-dimensional latent space can be considered a high-dimensional space for performing global optimization. General non-linear optimization in high-dimensional problems is an open problem. Most strategies assume that the optimization space can be simplified so that the optimization can be performed in a simpler or lower-dimensional space. For example, some authors [23,24,68] assume that the optimization space has some additive structure $f(x_1, x_2, ...) = f(x_1) + f(x_2) + f(...)$. Others perform only optimization in a local region [25] or assume the optimizer can be found in a lower-dimensional embedding [21,65,76]. For our setup, we hypothesize that a user's preferred sample can be found in a lower-dimensional embedding $S^d$ compared to the model latent space $S^D$, with dimensionality $d \ll D$, considering that some variables might be imperceptible for users not proficient in music or that certain regions of the search space might not capture harmony or proper musical structure.

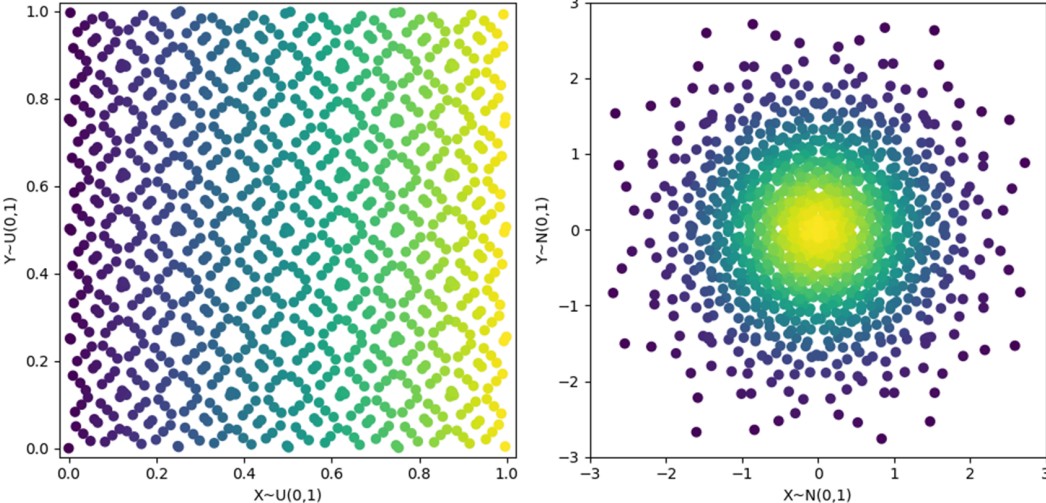

**Fig 4. Two-dimensional example of the Box-Muller transform.** The Sobol-sampled data (left) ends up following a Gaussian distribution (right) after being transformed. This allows Bayesian optimization to explore Gaussian spaces while respecting the prior of the distribution.

In this section, we present a couple of baselines for dimensionality reduction along with *random rotational embedding Bayesian optimization* (ROMBO), our proposed method for high-dimensional optimization. For the baselines, we first build a nonlinear decoder from $S^d$ to $S^D$ by training a variational autoencoder on top of the PowGAN model, which preserves the Gaussian distribution of the latent space. However, that increases the complexity of the model and the training process. Second, we use *random embedding Bayesian optimization* (REMBO) [21] as a baseline for high-dimensional Bayesian optimization. REMBO also builds a decoder to enable sampling on a lower-dimensional embedding, which in this case is a random linear projection. This guarantees finding a valid optimizer for the high-dimensional problem under certain conditions. REMBO is a fast and efficient method, but once again assumes a box-bounded input space, which conflicts with our Gaussian latent space. To solve this, we lastly introduce our REMBO-inspired algorithm, which we call *rotational random embedding Bayesian optimization* (ROMBO). The main advantages of ROMBO are its suitability for latent spaces with rotational symmetries, such as the Gaussian latent space of a GAN, and that its low-dimensional space bounds are simpler and fitter while still guaranteeing to find the optimum. Lastly, it is important to note that both REMBO and ROMBO are model-agnostic, meaning that they work with any model, not just PowGAN. Unlike a neural decoder, which requires changes in the architecture and re-training, the BO algorithms can be used out of the box.

**Neural decoder.** As a baseline to create a decoder from the low-dimensional space to the high-dimensional space, we propose a neural decoder extending the generator architecture of our music generation model following a *variational autoencoder* (VAE) methodology.

An autoencoder is composed of two networks: an encoder $E$ that reduces its input to fewer-dimensional features and a decoder $D$ that must reconstruct the original data from these features. Variational autoencoders extend this by turning the latent encoded feature points into latent distributions by imposing a Gaussian prior. This ultimately forms a Gaussian latent space which can later be sampled. This new, smaller latent space and the following layers that upsample it back to a piano roll make up our baseline. In our proposal, we salvage a pre-trained PowGAN and use the generator as $D$; the critic, sans the head, as $E$; and place a fully connected core that connects them and learns the latent space, including mean and standard deviation layers to enforce the Gaussian prior. In order to maintain the expressiveness of the GAN, we freeze the layers of PowGAN's generator and train just the encoder and core of the model. This is portrayed in Fig 5. At the end of training, we keep the neural encoder that has learned to map the feature space $S_d$ to PowGAN's latent space $S_D$.

In contrast to a GAN, where the two networks compete with each other, here $E$ and $D$ must both minimize a reconstruction loss $\mathcal{L}_R(x, D(E(x)))$. Common choices are the mean absolute error (MAE) and mean square error (MSE). However, these reconstruction errors are unfit for unbalanced, sparse, and heavily structured data, such as our piano rolls, where most values are silence and notes must follow specific patterns. We explore two alternatives for our training losses:

**Asymmetrical loss [77].** The first one is an asymmetrical loss ($\mathcal{L}_{AS}$), which focuses on element-wise correspondence while trying to compensate for note/silence imbalance, defined as

$$\mathcal{L}_{AS}(\Psi, \Psi') = \sum_{i,b,t,p}^{I,B,T,P} \psi_{ibtp} \cdot \mathcal{L}_+(\psi'_{ibtp}) + (1 - \psi_{ibtp}) \cdot \mathcal{L}_-(\psi'_{ibtp})$$

$$\mathcal{L}_+(x) = (1-x)^{\gamma_+} \log(x) \tag{7}$$

$$\mathcal{L}_-(x) = (x_m)^{\gamma_-} \log(x - x_m)$$

$$x_m = \max(x - m, 0)$$

where $\Psi$ and $\Psi' = D(E(\Psi))$ are the original and decoded piano rolls and $\psi_{ibtp}$ is the element at track $i$, bar $b$, timestep $t$ and pitch $p$ of a piano roll $\Psi$. $m$ is a negative clipping parameter, with values lower than $m$ being set to zero. $\gamma_-$ and $\gamma_+$, with $\gamma_- > \gamma_+$, are the asymmetric focus parameters that control how much negative predictions affect positive values and

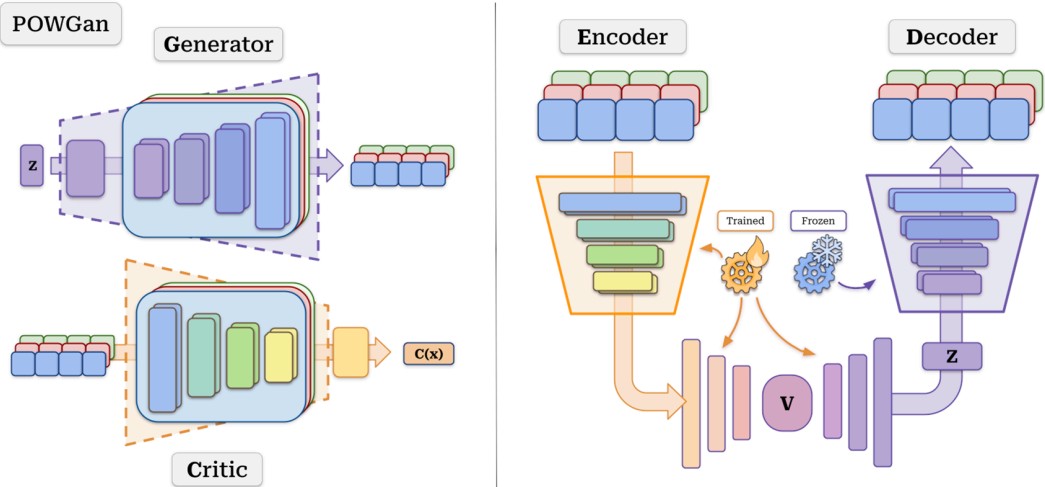

**Fig 5. Variational-autoencoder-inspired architecture.** We use PowGAN's beheaded critic C as encoder E, and generator G as decoder D. Features obtained through E are mapped to a low-dimensional Gaussian latent space ($S_d$) through fully connected layers. The feature vectors are then mapped back to PowGAN's high-dimensional latent space ($S_D$) and restored into piano rolls. All parts but PowGAN's generator are trained.

*vice versa*. Intuitively, this loss reduces the reward for easy true negatives, which prevents local minima such as complete silence (all zeros).

**Perceptual loss.** Element-wise losses can still lead to reconstructions with small errors that, perceptually, sound very different. This issue is tackled by using perceptual losses, which compare features, rather than single elements. One such example in the image domain is the structural similarity index measure (SSIM) [78], which compares patterns in pixel intensities. Thus, we designed a perceptual loss based on the previously commented musical features.

We first modify the EBR, as it can only output multiples of $\frac{1}{B}$ within the [0,1] interval for a single composition. We substitute it for an alternative, Silence Ratio (SR), calculated as

$$\text{SR}_i(\Psi) = 1 - \frac{1}{BTP} \sum_{b,t,p}^{B,T,P} \psi_{ibtp} \tag{8}$$

for each track *i* in a piano roll, which measures the percentage of silent elements rather than bars. Then, we formulate our musical feature loss as

$$\mathcal{L}_{MF}(\Psi, \Psi') = ||(\mathcal{F}_M(\Psi) - \mathcal{F}_M(\Psi'))||_2 \tag{9}$$

where $\mathcal{F}_M$ is a vector composed of the z-scores of the features calculated over the training data distribution for each of them. This includes SR for all instruments and TD and UPC for bass and guitar.

**KL divergence.** Whether we use $\mathcal{L}_{AS}$ or $\mathcal{L}_{MF}$ as $\mathcal{L}_R$, a Kullback-Leibler divergence (KL) term is always added to the loss to enforce a latent standard Gaussian distribution. The final loss of the architecture is

$$\mathcal{L} = \mathcal{L}_R(\Psi, \Psi') + \beta * \text{KL}(P_V || \mathcal{N}(\mathbf{0}, \mathbf{I_d})) \tag{10}$$

where $P_V$ is the latent space distribution, $\mathcal{N}(\mathbf{0}, \mathbf{I_d})$ is a *d*-dimensional standard Gaussian, and $\beta$ is a weighting term, which increases and resets regularly by applying KL annealing [79]. This practice prevents the model from ignoring the reconstruction error in favor of minimizing KL divergence, especially at the early stages of training.

**Random Embedding Bayesian Optimization.** Random Embedding Bayesian Optimization (REMBO) [21] works under the assumption that, for certain problems, most dimensions do not change the objective function. That is, the effective dimensionality of the problem is smaller than its input domain. One such example is shown in Fig 6, where the objective function is represented as a 2D function where there is only one important dimension, but we do not know which dimension is the important one at the beginning of optimization. Instead, we can build an embedded 1-dimensional subspace (represented as a red line in Fig 6) where we can perform optimization and we are guaranteed to find the optimum.

Formally, a problem with low effective dimensionality is defined as the following:

**Definition 1.** [21] *A function $f : \mathbb{R}^D \to \mathbb{R}$ is said to have effective dimensionality $d_e$, with $d_e \leq D$, if*

- *there exists a linear subspace $\mathcal{T}$ of dimension $d_e$ such that for all $\mathbf{x}_\top \in \mathcal{T} \subset \mathbb{R}^D$ and $\mathbf{x}_\perp \in \mathcal{T}^\perp \subset \mathbb{R}^D$, we have $f(\mathbf{x}_\top + \mathbf{x}_\perp) = f(\mathbf{x}_\top)$, where $\mathcal{T}^\perp$ denotes the orthogonal complement of $\mathcal{T}$.*
- *$d_e$ is the smallest integer with this property.*

*We call $\mathcal{T}$ the* **effective subspace** *of f and $\mathcal{T}^\perp$ the* **constant subspace**.

REMBO works by building a random linear projection $\mathbf{A} \in \mathbb{R}^{D \times d}$, where the elements of $\mathbf{A}$ are independently sampled from a standard normal distribution $\mathcal{N}(0, 1)$. Assuming that $d \geq d_e$, Theorem 2 of Wang et al. [21] shows that for any point in the original input space $\mathbf{x} \in \mathbb{R}^D$ there exists a point $\mathbf{y} \in \mathbb{R}^d$, such that $f(\mathbf{x}) = f(\mathbf{Ay})$. Therefore, for any optimizer $\mathbf{x}^*$ there exists a point $\mathbf{y}^*$ such that $f(\mathbf{x}^*) = f(\mathbf{Ay}^*)$.

A limitation of REMBO is how to define the search space for the lower-dimensional space to avoid missing any point. The authors show that, for any origin-centered box constraint space $\mathcal{X} \subset \mathbb{R}^D$ where $\mathcal{X}$ is rescaled to $[-1,1]^D$, and an effective dimensionality of $d_e$, then $||\mathbf{y}^*||_2 \leq \frac{\sqrt{d_e}}{\epsilon} ||\mathbf{x}_\top^*||_2 \leq \frac{\sqrt{d_e}}{\epsilon} \sqrt{d_e}$ with probability at least $1 - \epsilon$ [21]. In practice, the effective dimensionality $d_e$ is unknown, and we can only assume a dimension $d > d_e$. After empirical testing, they find $[-\sqrt{d}, \sqrt{d}]^d$ a suitable bound for the lower-dimensional space [21] which makes $\epsilon = \log(d)/\sqrt{d}$.

Our problem differs in that we want to find an optimizer $\mathbf{z}^*$ in a high-dimensional space $S^D$ with a standard Gaussian prior $\mathcal{N}(\mathbf{0}, \mathbf{I}^D)$. To take into account this prior, we would want our embedding $\mathbf{A}$ and the samples $\mathbf{y} \in S^d$ to ensure that $\mathbf{Ay}$ follows a multivariate Gaussian distribution and that its confidence bounds match those of $S^D$. We show how achieving

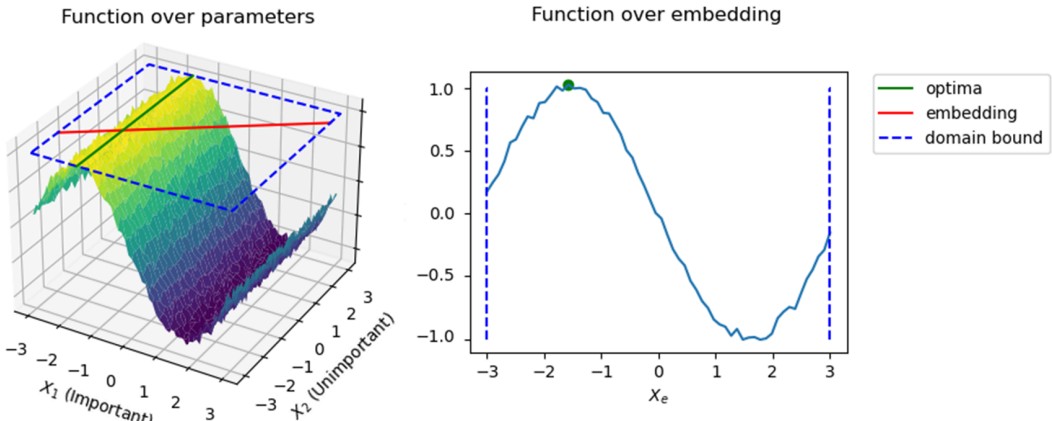

**Fig 6. Example of a function over a 2-dimensional domain and one effective dimension.** Searching over the 1-dimensional embedding is more efficient than searching over the whole domain.

this with REMBO and BM alone is not possible, and provide graphic examples in Fig 7 of their results and the desired embeddings, which we obtain through our own adapted method proposed below.

**REMBO(BM(y))** (Fig 7, left). Assuming our **y** is bounded within $[0,1]^d$, this method results in REMBO projecting the points into d-dimensional Gaussians in $S^D$. This satisfies the multivariate Gaussian requirement. However, due to the random scaling of the embedding, the distribution does not necessarily match the confidence bounds of $S^D$.

**BM(REMBO(y))** (Fig 7, middle). Embedding data within the interval $[-\sqrt{d}, \sqrt{d}]^d$ and applying the box constraint, as proposed in REMBO, projects **Ay** within $[-1,1]^D$. Re-scaling them to $(0,1)$ and applying Box-Muller forms spiral-like patterns. They provide no guarantee regarding the distribution of the data nor the confidence bounds, and sometimes create artifacts due to the box-constraint clipping, with many points being mapped either to infinity ($\ln(\epsilon)$ with some very small positive $\epsilon$) or the origin.

Since none of these approaches fulfill the requirements, we propose a novel methodology (Fig 7, right) that adapts REMBO to work as intended in Gaussian spaces.

## Random rotational embedding Bayesian optimization

Below, we explain how to adapt REMBO to improve its performance when working with Gaussian spaces and how to overcome some of its assumptions, like the box constraint.

Our proposed algorithm, which we call *random rotational embedding Bayesian optimization* (ROMBO), solves the scaling issue by redefining the embedding matrix so it acts as a rotation rather than a random projection. We provide our process for obtaining this new embedding and proof of its effectiveness below.

**Theorem 2.** *(based on Theorem 2 from Wang et al. [21]) Assume we are given a function $f : \mathbb{R}^D \to \mathbb{R}$ with effective dimensionality $d_e$ and a matrix $\mathbf{Q} \in \mathbb{R}^{D \times d}$, with $d \geq d_e$, whose columns form a random orthonormal basis for some d-dimensional subspace $\mathcal{Q} \subset \mathbb{R}^D$. Then, with probability 1, for any $\mathbf{z} \in \mathbb{R}^D$ there exists a $\mathbf{v} \in \mathbb{R}^d$ such that $f(\mathbf{z}) = f(\mathbf{Q}\mathbf{v})$.*

*Proof*: Following Wang et al. [21], we just need to prove that the matrix $\Phi^T\mathbf{Q}$, where $\Phi \in \mathbb{R}^{D \times d_e}$ is an orthonormal basis for $\mathcal{T}$, has rank $d_e$. The rest of the proof applies for any matrix $\mathbf{Q}$ that satisfies that. Both $\Phi$ and $\mathbf{Q}$ have orthonormal columns, and hence are full rank $d_e$ and $d$, respectively. Their product $\Phi^T\mathbf{Q}$ is a $d_e \times d$ matrix, with rank at most $d_e$. Since $\mathbf{Q}$ is randomly oriented relative to $\Phi$, $\Phi^T\mathbf{Q}$ will have rank $d_e$ with probability 1. □

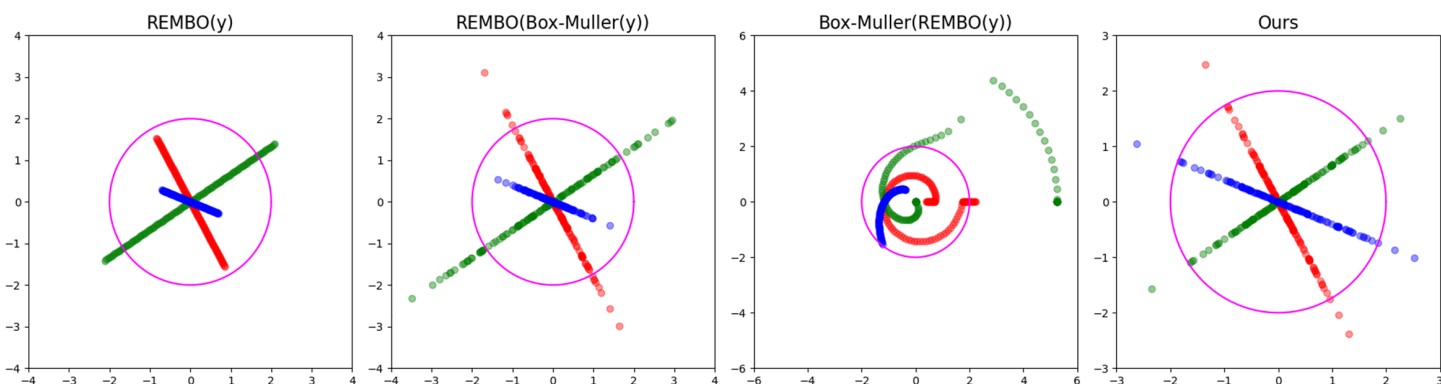

**Fig 7. Example embedding strategies.** The 1D data is embedded in a 2D space, where the 95% confidence bound of $\mathcal{N}(0, 1)$ is drawn in magenta. From left to right, data is embedded using REMBO on uniform data, REMBO on Gaussian data, Box-Muller after REMBO on uniform data, and our proposed method. The first two present scaling issues, and the third creates spiral patterns with undesirable artifacts. Our method embeds data as multivariate Gaussians with coincident confidence bounds with the standard distribution.

For our algorithm, we obtain $\mathbf{Q}$ from the *QR-decomposition* of $\mathbf{A}$, such that $\mathbf{A} = \mathbf{QR}$. We use the same strategy as REMBO and use a random $\mathbf{A}$ matrix whose entries are independent normal samples from $\mathcal{N}(0, 1)$. Then, as per the results from Steward [80], we can see that for any matrix $\mathbf{A}$ whose elements are independently normally distributed with mean zero and common variance $\sigma^2$, then $\mathbf{Q}$ is an orthonormal basis with random orientation. Although the result from Steward [80] is defined for full rank matrices $\mathbf{A_s} \in \mathbb{R}^{D \times D}$, with $\mathbf{A_s} = \mathbf{Q_s R_s}$, we can see that our matrix $\mathbf{Q}$ is just the first $d$ columns of $\mathbf{Q_s}$ representing the orthogonal basis for the column space of $\mathbf{A}$. Thus, if $\mathbf{Q_s}$ has random orientation, then $\mathbf{Q}$ has random orientation as well in the column space of $\mathbf{A}$.

Next, we need to consider the optimization problem in a compact space $\mathcal{Z} \in \mathbb{R}^D$ and how we can define the bounds in the embedding $\mathcal{V} \in \mathbb{R}^d$, which, as discussed before, is one of the limitations of REMBO.

**Theorem 3.** *(based on Theorem 3 from Wang et al. [21]) Suppose we want to optimize a function $f : \mathbb{R}^D \to \mathbb{R}$ with effective dimension $d^e \leq d$. Suppose further that the effective subspace $\mathcal{T}$ of $f$ is such that $\mathcal{T}$ is the span of $d_e$ basis vectors, and let $\mathbf{z}_\top^* \in \mathcal{T}$ be an optimizer of $f$ inside $\mathcal{T}$. If $\mathbf{Q}$ is a $D \times d$ matrix such that its columns form a random orthonormal basis for $\mathbb{R}^d$, there exists an optimizer $\mathbf{v}^* \in \mathbb{R}^d$ such that $f(\mathbf{Qv}^*) = f(\mathbf{z}_\top^*)$ and $||\mathbf{v}^*||_2 = ||\mathbf{z}_\top^*||_2$.*

*Proof*: In this case, our proof is simpler than Theorem 3 [21]. Since $\mathbf{Q}$ is an orthonormal basis, $\mathbf{Qv}^*$ is a rotation of $\mathbf{v}^*$ in $S^D$, as all $d$ singular values of $\mathbf{Q}$ are 1. Therefore, $||\mathbf{v}^*||_2 = ||\mathbf{Qv}^*||_2 = ||\mathbf{z}^*||_2$. $\square$

This result is quite important. Contrary to REMBO, our projection matrix $\mathbf{Q}$ does not change the size of our exploration space. This is intuitive when we consider that REMBO builds its input space in a hypercube. Meanwhile, our input space is a hypersphere thanks to the rotational symmetry of the latent space as a Gaussian distribution, and the Box-Muller transformation. Therefore, we can maintain the same constant bounds independently of the effective dimension, and therefore, we can find tight bounds even if the effective dimension is unknown. As shown in Fig 7, our method preserves the bounds of the search and input spaces.

## Pipeline evaluation

### Music generation model

The model architecture follows the schema displayed in Fig 2. As discussed, our final piano rolls have three instruments instead of five. The temporal resolution of our output is also coarser, going down from 48 timesteps per bar to 16. The final output of our model is a piano roll $\Psi \in [0, 1]^{3 \times 4 \times 16 \times 72}$. We provide a layer breakdown of the generator in Table 1.

For the training data, each song is divided into 4-bar fragments, which are then sampled uniformly up to a maximum of 18 fragments per song. We ensure that removing the piano and strings does not accidentally create silent fragments, though we maintain them if they were already silent, even with all instruments. Our final training dataset is made of 57.201 fragments. For evaluation, we compare the musical features of the generated songs and the dataset used for training. We measure the EBR, UPC, and TD of a new dataset of 64000 fragments (2000 batches of 32) generated by sampling

Table 1. **Layer breakdown of PowGAN's generator.** 'Shared' layers are common for all instruments, while 'Private' ones are replicated per instrument. Private pitch-first and time-first layers are parallel, and they end up merging with a 1x1 convolution.

| | Kernel | Stride | Output | Channels |
|---|---|---|---|---|
| **Shared network** | (4,1,1) | (4,1,1) | (4,1,1) | 256 |
| | (1,4,3) | (1,4,3) | (4,4,3) | 128 |
| | (1,2,3) | (1,2,2) | (4,8,6) | 64 |
| **Private pitch-first** | (1,1,12) | (1,1,12) | (4,8,72) | 16 |
| | (1,2,1) | (1,2,1) | (4,16,72) | 1 |
| **Private time-first** | (1,2,1) | (1,2,1) | (4,16,6) | 16 |
| | (1,1,12) | (1,1,12) | (4,16,72) | 1 |
| **Private merge** | (1,1,1) | (1,1,1) | (4,16,72) | 1 |

PowGAN's high-dimensional Gaussian latent space. We report the metrics obtained by the PowGAN generations in Table 2, and provide the ones reported by MuseGAN [26] as reference. The overall absolute error between our dataset and our model for each of these metrics is similar in scale to the errors reported for MuseGAN, and even improves them in the case of UPCs and EBR for the drum track. With these results, we deem the training as correct and use PowGAN as our music generation model for the optimization tests.

The code for training and evaluation was implemented using PyTorch and based on MuseGAN's implementation. We train the model for 25000 steps, with a batch size of 32 and using an Adam optimizer with $\beta_1 = 0.6$ and $\beta_2 = 0.9$ and otherwise PyTorch default parameters. The generator and critic each use a different optimizer, and at each step, the critic performs 5 forward passes for every forward pass of the generator. For the first 10 steps, these passes are increased to 50, to give the critic a head start. The training was performed on a free Google Colaboratory environment equipped with an Intel Xeon CPU with 2 vCPUs and 13GB of RAM, as well as an NVIDIA Tesla T4 GPU with 16GB of VRAM. The whole execution of the notebook takes about an hour to complete under these conditions.

## High-dimensional optimization

The neural decoders are trained using the dataset generated with PowGAN, which is split in a 90/10 ratio for the train and test sets. We use this and not PowGAN's training data because a GAN generator tends not to replicate the individual data points used to train it [81]. Our decoder is a frozen pre-trained generator, so we cannot confirm whether it can generate the exact pieces used to train PowGAN or not. With a training set made of its generations, we ensure the decoder can replicate them once again.

We use both $\mathcal{L}_{AS}$ and $\mathcal{L}_{MF}$ to train two neural decoders, referred as $ND_{AS}$ and $ND_{MF}$ below. The training was performed on a 12th Gen Intel Core™ i7-12700K CPU with 32GB of RAM and an NVIDIA GeForce RTX 4090 GPU with 24GB of VRAM. We train for 10 epochs and a batch size of 32. The execution takes between fifteen and thirty minutes under these conditions, depending on the loss used for training. REMBO and ROMBO do not need additional training, as discussed earlier. To run the experiments, we develop an application based on the BO library BayesOpt [73]. We use the Python interface for BayesOpt, which provides a wrapper for custom optimization problems.

We design a series of experiments to test the efficacy of the trained neural decoders, REMBO, and ROMBO. The objective of each experiment is to recreate a song generated by forward-passing a point randomly sampled from $S^D$ through the PowGAN generator. This song represents a hypothetical subject's favorite song, which acts as the target. A total of 20 targets are generated, which are the same for all decoders.

Our objective function starts by taking a query point from our $S^d = [0, 1]^d$. For REMBO, it rescales it to the recommended $[-\sqrt{d}, \sqrt{d}]^d$ interval. For the others, it performs the Box-Muller transform into $\mathcal{N}(\mathbf{0}, \mathbf{I}^d)$. The resultant query is projected onto $S^D$ through the selected decoder, and the generator produces the resultant song. The loss (either $\mathcal{L}_{AS}$ or $\mathcal{L}_{MF}$)

**Table 2.** **Metric comparison between the used dataset and the closest values reported in evaluation, both for the MuseGAN baseline and Pow-GAN.** Values in the 'Model' row are better the closer to the 'Dataset' row. Notice that both datasets show different metrics due to random sampling, track trimming, and sample rejection, but the error still shows a similar scale, which suggests correct training. For the columns, 'D', 'B', and 'G' correspond to Drums, Bass, and Guitar.

| | | EBR (%) | | | UPC | | TD |
|---|---|---|---|---|---|---|---|
| | | D | G | B | G | B | G-B |
| MuseGAN | Dataset | 8.06 | 19.4 | 8.06 | 3.08 | 1.71 | 1.57 |
| | Model | 2.33 | 18.3 | 6.59 | 3.69 | 1.53 | 1.56 |
| | Abs. Error | 5.73 | 1.10 | 1.47 | 0.61 | 0.18 | 0.01 |
| PowGAN | Dataset | 6.40 | 20.07 | 8.74 | 3.43 | 2.25 | 1.27 |
| | Model | 8.50 | 18.75 | 6.53 | 3.36 | 2.14 | 1.32 |
| | Abs. Error | 2.10 | 1.32 | 2.21 | 0.07 | 0.11 | 0.05 |

between this song and the target is the function output. For $\mathcal{L}_{AS}$ we chose $\gamma_+ = 4, \gamma_- = 16$ and $m = 0.05$ as our hyperparameters.

We select $d = 10$ for our latent space dimensionality. All experiments use the 5/2 Matèrn kernel with ARD and the Expected Improvement acquisition function. For REMBO and ROMBO, all trials were done with a single random embedding per target, and the embedding for ROMBO was the **Q** matrix obtained from the QR-decomposition of the **A** matrix used for REMBO for that same target. For completeness, we also tested standard BO with no adaptation for high dimensionality, and a naïve approach where we don't perform optimization. This last method randomly queries the high-dimensional latent space 70 times and saves its best result, but the model receives no feedback. All other methods start each trial with Sobol sampling, collecting 10 data points, and the optimization budget after initialization is set to 60 samples.

The results of the automatic trials, collected in Figs 8 and 9 Table 3, show the evolution and final average values of each loss function through optimization. The results for standard BO have been omitted from these for clarity, as its performance was worse than random due to the high dimensionality of the sampling space.

Firstly, the random baseline obtains far higher losses, especially $\mathcal{L}_{MF}$, than ROMBO and REMBO. This shows that high-dimensional BO algorithms help navigate the latent space of the model. ROMBO outperforms, on average, both REMBO and the neural encoders. This is true for both $\mathcal{L}_{AS}$ and $\mathcal{L}_{MF}$ even when optimizing with a metric and evaluating the other. ROMBO shows an early advantage that persists throughout optimization, suggesting better exploration, as intended when formulating the method.

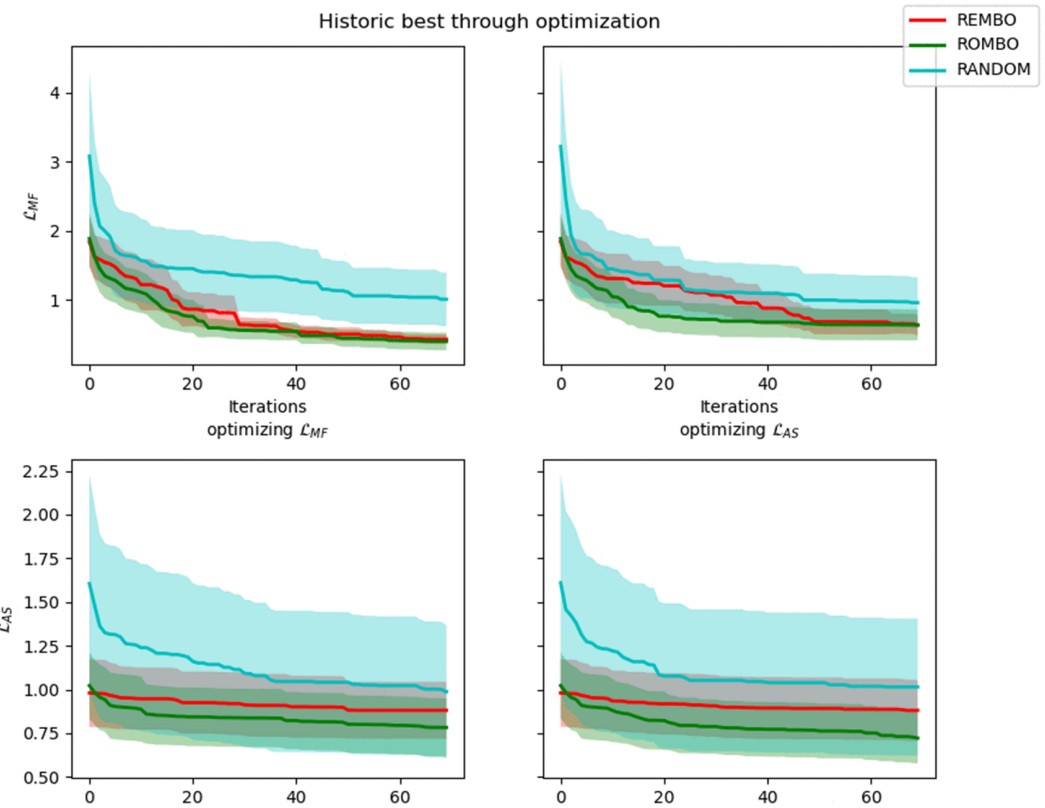

**Fig 8**. **Average historic best (with 95% confidence bounds) showcasing the evolution of $\mathcal{L}_{MF}$ (top) and $\mathcal{L}_{AS}$ (bottom) during optimization of $\mathcal{L}_{MF}$ (left) and $\mathcal{L}_{AS}$ (right) for ROMBO (green), REMBO (red), and naïve random search (cyan).** ROMBO outperforms both in all metrics and targets.

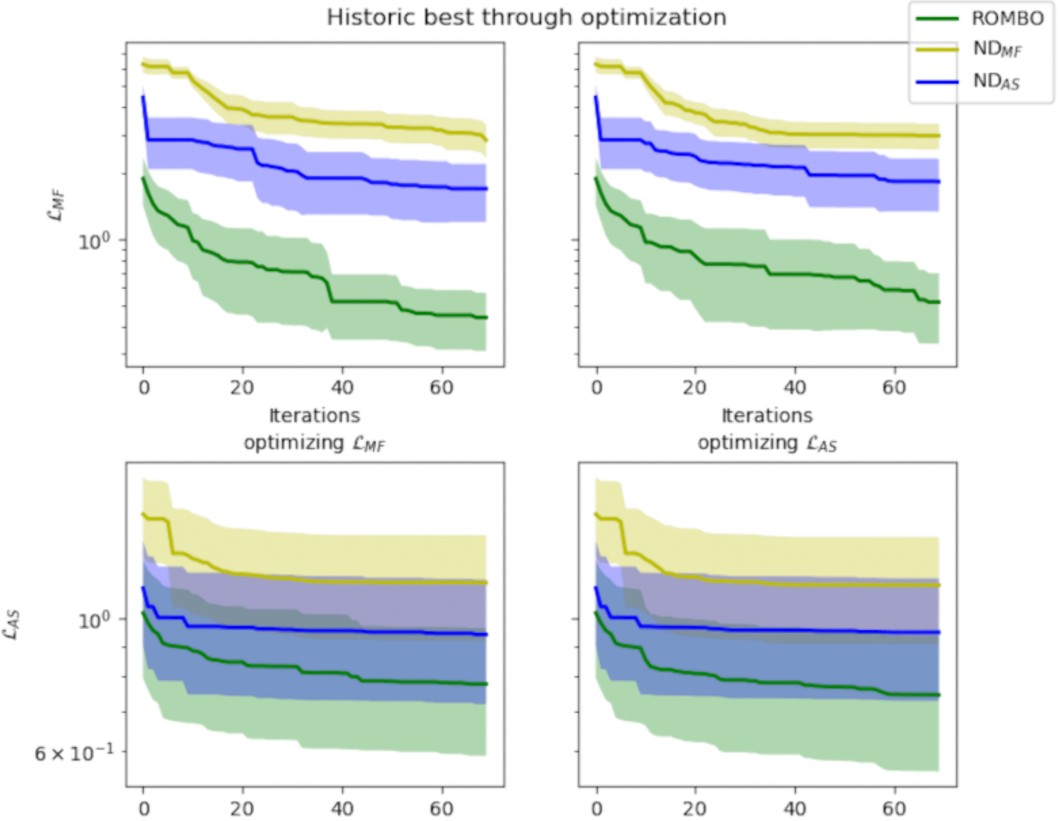

**Fig 9**. Average historic best (with 95% confidence bounds, plotted in logscale) showcasing the evolution of $\mathcal{L}_{MF}$ **(top)** and $\mathcal{L}_{AS}$ **(bottom)** during optimization of $\mathcal{L}_{MF}$ **(left)** and $\mathcal{L}_{AS}$ **(right)** for ROMBO **(green)** and the NDs trained minimizing $\mathcal{L}_{MF}$ **(yellow)** and $\mathcal{L}_{AS}$ **(blue)**.

**Table 3**. **Final average loss after optimization for each decoder and loss combination.** ROMBO outperforms REMBO, naive random search, and both NDs in both metrics when optimizing either of the two loss functions.

| Optimizing: | With decoder: | Obtained metric | |
|---|---|---|---|
| | | $\mathcal{L}_{MF}$ | $\mathcal{L}_{AS}$ |
| $\mathcal{L}_{MF}$ | $ND_{MF}$ | 2.8240 | 1.1502 |
| | $ND_{AS}$ | 1.6969 | 0.9410 |
| | Random | 1.0043 | 0.9868 |
| | REMBO | 0.5728 | 0.9185 |
| | ROMBO | **0.4397** | **0.7762** |
| $\mathcal{L}_{AS}$ | $ND_{MF}$ | 2.9550 | 1.1400 |
| | $ND_{AS}$ | 1.8268 | 0.9488 |
| | Random | 0.9558 | 1.0149 |
| | REMBO | 0.7437 | 0.9010 |
| | ROMBO | **0.5164** | **0.7444** |

However, random search beats the neural encoders, the one exception being the $\mathcal{L}_{AS}$ obtained by $ND_{AS}$. This suggests that the models fail to capture the features of the final songs in a reduced learned space. The curves of the neural decoders also offer additional insight into their behavior. Despite being trained to minimize their respective losses, the one trained on $\mathcal{L}_{MF}$ is significantly worse at both. This may be due to the model itself not being complex enough or the loss not being fit for gradient-descent-based strategies. As for the $ND_{AS}$, its performance is closer to ROMBO, though it still

doesn't reach the same values, and the latter has less variance in the results. Both metrics follow similar curves during optimization, with slight improvements for the optimized metric at each run.

The results show that ROMBO is the best method out of those we could develop and are proof of our method's effectiveness. All further tests are carried out using ROMBO as our decoder.

## Experiment design and procedure

We conclude our work with a human-driven test.

### Experimental design

We switch our evaluation method to human judgment, a numerical evaluation from 0 to 10, and they are asked to try and find a favorite generation. Participants were recruited in February 2025. The only requirement for enrollment was a healthy hearing. Musical experience was not addressed. A total of sixteen people enrolled, four female and twelve male participants between the ages of 23 and 43 (average 27.44, standard deviation 5.82), who were divided into two groups of eight (from now on the 'Bayesopt' and 'Random' groups). The first group uses the whole pipeline with ROMBO, and the second listens to random samples from PowGAN. Participants are not told this information. The objective is to evaluate the optimization's effectiveness compared to randomly asking the model to generate pieces and to see whether users think the sample refinement process is useful, regardless of the selected method.

### Ethics statement

The research process was conducted according to the ethical recommendations of the Declaration of Helsinki and was approved by both our institution's data protection unit and the Aragón Autonomous Community Research Ethics Committee (CEICA), which supervises fairness in research projects and the use of biological and personal data. Our protocol is non-invasive and involves the gathering of opinion data only, with no risk for any participant, who gave their informed written consent right after explaining the experiment. The consent allowed the abandonment of the experiment, and all data was analyzed anonymously.

### Procedure

Once the volunteer has given their written consent, the supervisor tells them the following pieces of information:

- They will be listening to several computer-generated, 10-second-long musical compositions. The objective is to find the song that is best suited to their taste among the possible generations.
- They will be giving a score to each of the compositions, from 0 to 10, and with 0.1 precision. Scores are said aloud to the researcher. Once they confirm their decision, the scores can't be changed. If they give the same highest score twice, the latter will be considered better.
- They can ask the supervisor to replay each composition once, before giving the score, if they need to. They can also ask to replay the one they gave the current best score to and be reminded of what the score was.
- The experiment lasts for approximately 30 minutes, accounting for short breaks every few songs. The supervisor will tell them when a break starts and stops.

Then, the supervisor generates and plays two random songs. The user is told not to give a score for these songs. This serves two purposes: First, letting the participant know how the generated songs sound, to control expectations. We transform our output piano rolls to MIDI and play them using standard sound fonts. Hearing some samples in advance gives users a preamble to palliate common biases of opinion problems, such as drift and anchoring. Second, giving them

the chance to adjust the volume. The user is instructed to turn it up enough so they can clearly hear the music, but not so much as to be uncomfortable for them to listen to.

We choose the experiments' parameters taking into account the limitations of real users, which mainly involve evaluation time, capping the initial dataset size and iteration budget to a 30-minute time limit. The final number of observations is 22 samples using Sobol and 42 samples using optimization, for a total of 64 samples per participant. Compared to the automatic trials (10 Sobol samples and 60 acquired samples), this gives volunteers more chances to hear different kinds of compositions, potentially finding one that caters to their taste. It is also a way to prepare for noisy observations, as automatic trials used an exact loss, but we cannot expect the same for human evaluation. The breaks are distributed uniformly along the duration of the experiment, with two thirty-second breaks when reaching 16 and 48 evaluations, and a one-minute break when reaching 32 of them. These breaks both reduce the possible fatigue of the user and reset the hearing, so evaluations are not that conditioned on how long users have been listening to music without stopping.

Then, after completing the optimization process, volunteers are introduced to a survey stating four affirmations about the model's general performance and the optimization process. All statements can be answered using Likert scale answers: "Strongly disagree", "Disagree", "Neither agree nor disagree" (onward "No preference", for short), "Agree", and "Strongly agree". Users are asked to tell how much they agree with the following points:

- Q1: "Overall, I like the generated songs"
- Q2: "I like the song I gave the best score to"
- Q3: "I like the song I gave the best score to better than the first song I listened to"
- Q4: "After listening to all the songs, I am alright with the scores I gave to the best and first songs"

Question Q1 asks about the general performance of the composition model. Questions Q2 and Q3 evaluate the need for an optimization pipeline. Finally, question Q4 asks about the users' ability to act as judges, since they are not given a chance to correct themselves and may not agree with their own judgment after the experiment ends.

## Results

For the response to the final survey, Fig 10 shows the distribution of answers for all 16 participants. Most participants answer either neutrally or negatively to Q1, meaning they do not explicitly like the generated songs in general. However, both groups agree that they like their final selected piece, and most times they like it better than the first one they heard (Q2 and Q3), meaning that users are not willing to settle for the first things they obtain from a generative model and appreciate customization of deep learning models to individual tastes. Finally, most users agree with the grades they have given during the experiment (Q4), which supports our experimental procedure and our choice of scalar grades as a goodness function.

We next compare the musical features obtained by each of our volunteers' final pieces and the model's average metrics. For this analysis, we use the same metrics used to evaluate the training of the composition model: UPC for guitar and bass, and the TD between them. SR is omitted for brevity's sake, as it is stable across all generated compositions, with small variability in the whole latent space. Fig 11 and Tables 4 and 5 show a per-user display of the obtained metrics.

Overall, the metrics show significant variations among different users. The Random group has a closer-to-average UPC for the guitar track, but a much larger one for the bass track. This also has an impact on the TD, as two harmonic tracks (both playing chords) tend to have lower distances than a harmonic track and a melody track (one playing chords and the other single notes), which is the common behavior of the guitar and bass tracks [26]. Considering that for the Random group exploration was not guided by feedback, unlike for the Bayesopt group, this may be an indicator of the optimization pipeline helping to avoid irregular regions of the latent space. Meanwhile, only one corner case is found among the Bayesopt group: subject 4. In their experiment, they gave high scores to compositions that silenced both the bass and guitar tracks, finding the optimum in a drums-only composition. The model's probability to do so on its own, without human

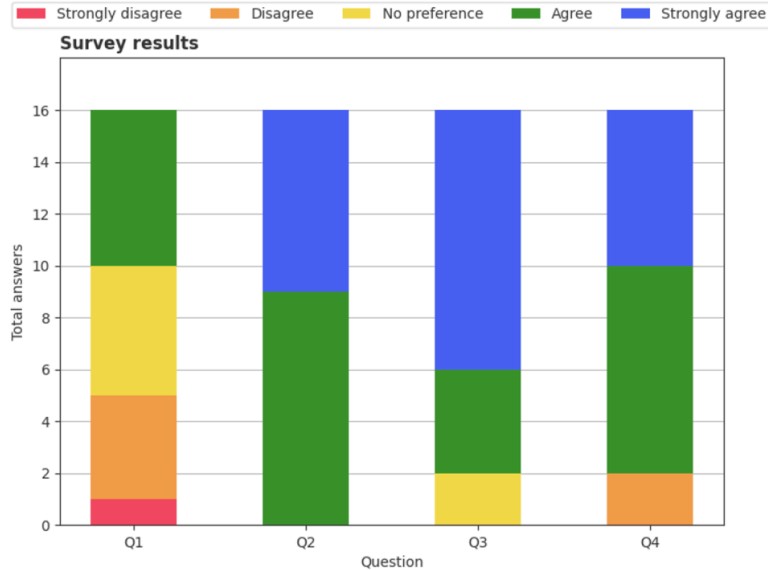

**Fig 10**. **Column charts showcasing the results of the Likert survey.** All questions show overall positive responses. Though most users do not explicitly like most of the pieces (Q1), they always like the one they graded best (Q2), in most cases better than the first one (Q3), and they are alright with their grades (Q4).

guidance, lies at 0.9%. In this case, the irregular behavior was intentional, as the user rewarded the first drum solo they encountered, and the system started generating more until the end of the experiment.

As a final test with both groups, we compared the impact of Bayesian optimization during the duration of the experiment by looking at the grades' evolution with and without the optimization running. Our measurements focus on the experience of users when using the pipeline, emphasizing efficiency. Optimization should help users find preferred compositions faster and more often. Also, exploration should be guided and avoid uninteresting regions within possibility. Specifically, we retrieve the number of maximum score updates along the whole optimization and define two metrics: no-new-max streaks and low-grade streaks. No-new-max streaks are iterations in a row without a new maximum, whereas Low-grade streaks are periods where the average of the last three scores is below the average of the maximum and minimum values so far. The first one measures how often users update their best graded composition, and the second measures how long they spend giving sub-par grades in a row. The collected metrics are showcased in Table 6.

These metrics show how the users in the Bayesopt group have shorter no-new-max streaks and more best score updates overall. Their low-grade streaks are also shorter. This suggests that optimization may indeed help avoid uninteresting areas of the search space. The difference in best score updates stays consistent over the optimization process, as showcased in Fig 12, including the initial budget acquisition. This emphasizes the inefficiency of high-dimensional space exploration and suggests that ROMBO is an effective way to reduce dimensionality without hindering performance.

After a qualitative analysis of the experiment results and the volunteers' behavior and comments throughout its duration, we gathered some additional findings. First, though they are allowed to replay the fragments, few of the participants used this functionality. Even at the final survey, when asked if they needed to replay the first fragment (which they listened to 30 minutes earlier), some said they remembered it. We take this as an indicator of memorization for comparison not being an issue, with early samples still providing information later on. Second, though allowed to use 0.1 precision, the general tendency is to use rounded grades or 0.5 precision. This particular behavior is of special importance for the last part of the second clause of the experiment stated above ("if they give the same highest score twice, the latter will be considered better"), as it is something users exploit. If the grade difference between the best fragment candidates is small,

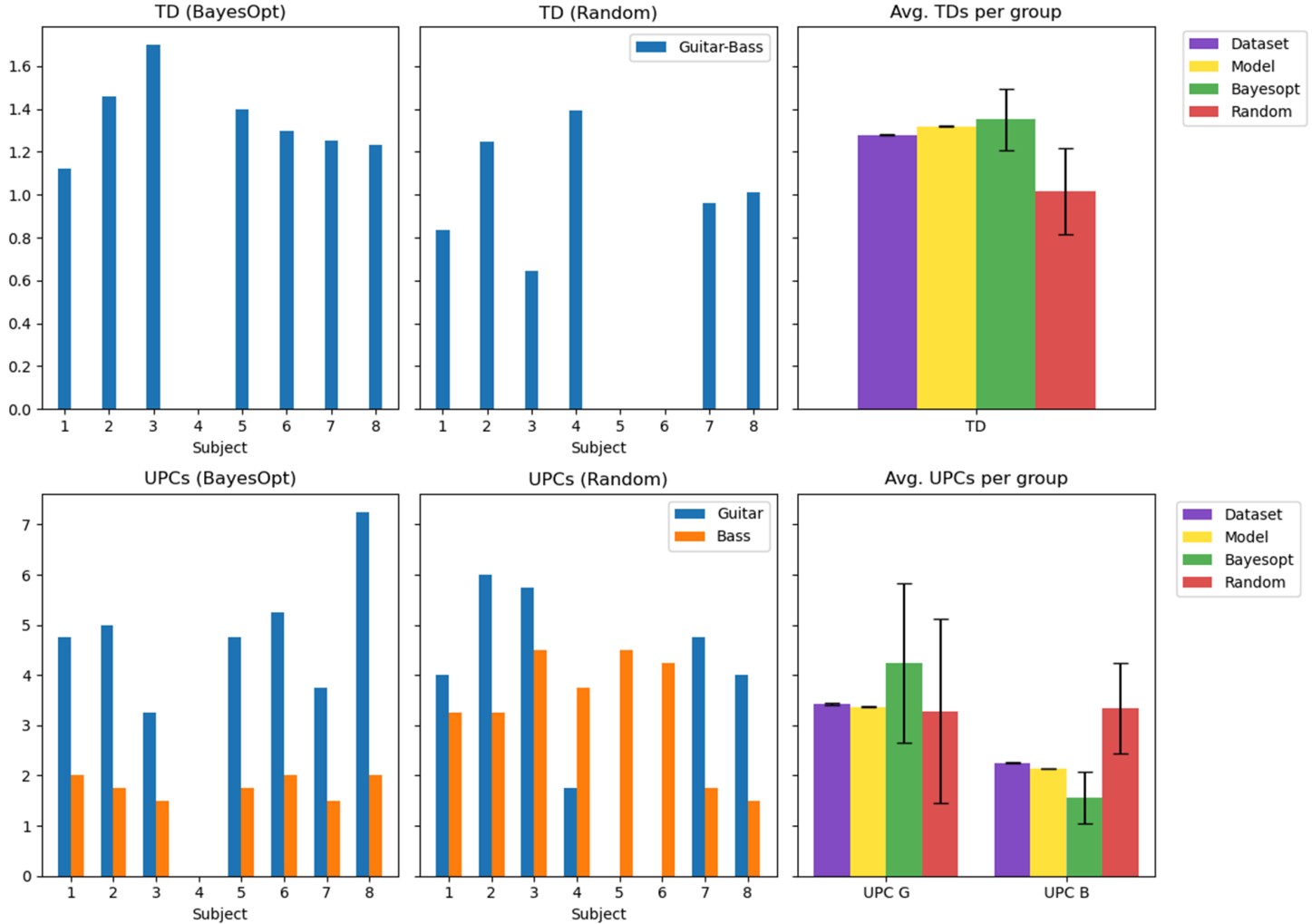

**Fig 11**. **Per-user bar chart of UPCs and TD for each of the participants' best-graded compositions, and average results (with 95% confidence bounds) for the dataset, the model, and the Bayesopt and Random groups.** The Bayesopt group shows a close TD and a good Guitar-Bass balance for UPCs, while the Random group tends to select songs with high UPC in the bass, hinting at exploration around irregular regions of the distribution.

**Table 4**. **Metrics of the Bayesopt group participants' final selected songs.** UPCs for guitar (G) and bass (B) are consistently above and below average, respectively, and TD is slightly higher.

|  | Dataset avg. | Model avg. | Group avg. | S1 | S2 | S3 | S4 | S5 | S6 | S7 | S8 |
|---|---|---|---|---|---|---|---|---|---|---|---|
| **UPC (B)** | 2.25 | 2.14 | 1.56 | 2.00 | 1.75 | 1.25 | 0.00 | 1.75 | 2.00 | 1.50 | 2.00 |
| **UPC (G)** | 3.43 | 3.36 | 4.25 | 4.75 | 5.00 | 3.25 | 0.00 | 4.75 | 5.25 | 3.75 | 7.25 |
| **TD (B-G)** | 1.27 | 1.32 | 1.35 | 1.12 | 1.45 | 1.69 | N/A | 1.39 | 1.29 | 1.25 | 1.23 |

instead of upping the ante by 0.1, most users' preferred solution is to keep the round score and override it. We make sure that this does not greatly impact our pipeline's performance, as Fig 12 shows, as even with strictly higher scores, our optimization group still obtains more updates for their best graded compositions. This may be due to users preferring to keep a reduced collection of distinct given scores to facilitate keeping track of the fragments they have already graded. Overall,

**Table 5**. **Metrics of the Random group participants' final selected songs.** The Guitar track reports an average UPC close to the model mean, but UPCs for the Bass track are greatly above average. TD is consistently lower than the model's average.

| | Dataset avg. | Model avg. | Group avg. | RS1 | RS2 | RS3 | RS4 | RS5 | RS6 | RS7 | RS8 |
|---|---|---|---|---|---|---|---|---|---|---|---|
| UPC (B) | 2.25 | 2.14 | 3.34 | 3.25 | 3.25 | 4.50 | 3.75 | 1.75 | 4.5 | 1.75 | 1.50 |
| UPC (G) | 3.43 | 3.36 | 3.28 | 4.00 | 6.00 | 5.75 | 1.75 | 0.00 | 0.00 | 4.75 | 4.00 |
| TD (B-G) | 1.27 | 1.32 | 1.01 | 0.84 | 1.25 | 0.64 | 1.39 | N/A | N/A | 0.96 | 1.01 |

**Table 6**. **Metrics for both our Bayesopt and Random groups.** All metrics are averaged over the subjects of each group. "No-new-max streaks" are periods without a new maximum in their grades. "Low-grade streaks" are periods with a 3-iteration rolling average below the mean of the maximum and minimum values up to the current iteration. The 'Difference' column shows how much the Bayesopt group metrics improve with respect to the Random group.

| Group | | Bayesopt | Random | Difference |
|---|---|---|---|---|
| # of updates (↑) | Avg. | 7.00 | 5.00 | +2.00 |
| No-new-max streak (↓) | Avg. | 8.29 | 9.96 | −1.67 |
| | Median | 2.75 | 3.93 | −1.18 |
| | Longest | 34.62 | 36.50 | −1.88 |
| Low-grade streak (↓) | Avg. | 2.89 | 3.52 | −0.63 |
| | Median | 2.19 | 2.56 | −0.37 |
| | Longest | 7.75 | 8.75 | −1.00 |

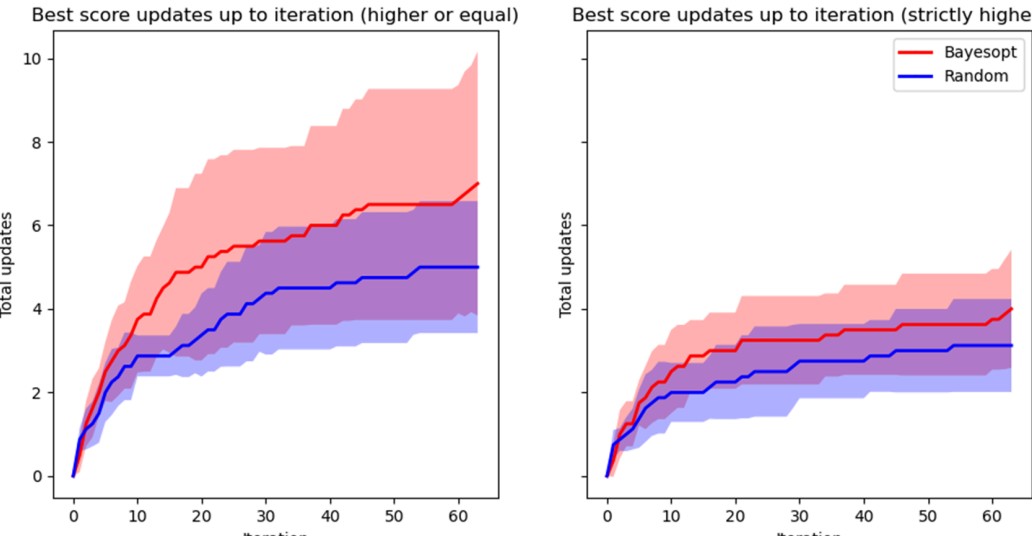

**Fig 12**. **Evolution of the average maximum value update number (with 95% confidence bounds) throughout optimization for the Bayesopt and Random groups.** The left side shows the experiment's setup, with equal values given later being considered new maxima. The right side shows evolution using strictly higher scores, for analysis completion. In both cases, the optimization group shows a consistent advantage.

results show that the optimization process helps sample better candidates than randomly querying the model, and users seem to find the pipeline useful and quickly gain proficiency with it.

## Conclusion and future work

We present random rotational embedding Bayesian optimization (ROMBO), a novel Bayesian optimization algorithm for high-dimensional optimization in Gaussian search spaces. In our theoretical analysis and numerical experiments, we

show that ROMBO outperforms another high-dimensional Bayesian optimization baseline (REMBO), which assumes box-bounded search spaces, when optimizing over the Gaussian latent space of the model. This shows that, when working with these Gaussian high-dimensional spaces, it is important to ensure the embedded data preserves the sampling distribution of the high-dimensional space. Our algorithm also outperforms dimensionality reduction with neural autoencoders.

We validate ROMBO in a personalized music generation system. We replace the standard random sample for the generative process with a specifically selected sample that best caters to user preferences. For that, we build an optimization pipeline that iteratively generates pieces of music and receives user ratings. We perform online optimization using ROMBO to find the optimal sample seed. Our pipeline is efficient in two ways. First, ROMBO is a Bayesian optimization method, known for its sample efficiency among global optimization methods. This reduces the number of user pieces of music generated and rated needed to find the desired one. Second, by using a lightweight generative model (PowGAN), based on the gold-standard model MuseGAN. Using PowGAN we are capable of generating piano rolls for drums, guitar, and bass. To our knowledge, this is the first work to apply Bayesian optimization for the complex case of several polyphonic instrument tracks. That said, one key feature of our pipeline is that ROMBO, as any Bayesian optimization algorithm, is a black-box optimization method. Therefore, it is *agnostic* to the generative model used or the type of ratings presented by the humans, as soon as they are consistent.

We evaluate our music generation pipeline with ROMBO in a pilot user study with 16 subjects. The results reveal that Bayesian optimization, and ROMBO in particular, are effective means to personalize generative model outcomes for humans. We compare a group of subjects using ROMBO and a control group using random samples with no optimization or feedback. The results of the Bayesopt group show how ROMBO can find more pieces with a high rating. ROMBO also guides exploration to avoid local optima, still producing diverse results in the initial stages. We show how the pieces generated by ROMBO are, on average, more natural and closer to actual human compositions compared to the random samples, while still allowing users to deviate from that. The final survey results reveal that, while the users liked the quality of the top-rated songs, the rest of the songs got a lukewarm reaction. We believe that this is mainly a limitation of the model. While composition-wise it is very capable, a lightweight MIDI generator has limited power for rendering the final audio, as it depends on standard libraries.

Future lines of work may focus on comparing to state-of-the-art solutions for music encoding, and how BO can be implemented along with them, thanks to our pipeline being model agnostic. In addition, other high-dimensional BO methods, such as sparse priors, could be combined with our method to further improve performance. Regarding the human feedback experiments, future work can expand this line to reach more robust conclusions. Our experiment confirmed the findings of our simulated trials, but had a limited number of participants. However, with our algorithm developed and tested, further research can continue this study. New studies can address potential confounding variables such as gender or musical background. Moreover, a study researching why users like or dislike certain pieces, perhaps leveraging architectures not contemplated in this study, such as large language models, can be of interest. Additionally, the pipeline can also be evaluated with other state-of-the-art architectures for music generation or even other modalities, such as text, image, video, etc. As a piece of software, the pipeline can also be integrated into music editing and completion workflows, or musical therapy apps. Rather than generating music from scratch, the human aspect of this paper can be enhanced by increasing the interaction with the model. Finally, we intend to apply ROMBO in other applications beyond generative models where Bayesian optimization is present, such as robotics, reinforcement learning, automatic machine learning, and simulation.

## Author contributions

**Conceptualization:** Miguel Marcos, Lorenzo Mur-Labadia.

**Data curation:** Miguel Marcos.

**Formal analysis:** Miguel Marcos.

**Funding acquisition:** Ruben Martinez-Cantin.

**Investigation:** Miguel Marcos.

**Methodology:** Miguel Marcos.

**Project administration:** Miguel Marcos.

**Resources:** Miguel Marcos, Ruben Martinez-Cantin.

**Software:** Miguel Marcos.

**Supervision:** Lorenzo Mur-Labadia, Ruben Martinez-Cantin.

**Validation:** Miguel Marcos.

**Visualization:** Miguel Marcos.

**Writing – original draft:** Miguel Marcos, Lorenzo Mur-Labadia, Ruben Martinez-Cantin.

**Writing – review & editing:** Miguel Marcos, Lorenzo Mur-Labadia, Ruben Martinez-Cantin.

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
