## [Decision Letter · Decision Letter 0]

15 Aug 2025

PONE-D-25-27090

Random rotational embedding Bayesian optimization for human-in-the-loop personalized music generation

PLOS ONE

Dear Dr. Marcos,

Thank you for submitting your manuscript to PLOS ONE. After careful consideration, we feel that it has merit but does not fully meet PLOS ONE’s publication criteria as it currently stands. Therefore, we invite you to submit a revised version of the manuscript that addresses the points raised during the review process.

We look forward to receiving your revised manuscript.

Kind regards,

Tara Rajendran, MBBS, Ph.D., MFA

Guest Editor

PLOS ONE

Journal Requirements:

[This research was funded by Spanish government projects PID2021-125209OB-I00 (https://www.aei.gob.es/) and the Aragon Government DGA T45_23R (https://www.aragon.es/).].

Reviewers' comments:

Reviewer's Responses to Questions

**Comments to the Author**

1. Is the manuscript technically sound, and do the data support the conclusions?

Reviewer #1: Partly

Reviewer #2: Yes

Reviewer #3: No

Reviewer #4: Yes

2. Has the statistical analysis been performed appropriately and rigorously?

Reviewer #1: I Don't Know

Reviewer #2: Yes

Reviewer #3: Yes

Reviewer #4: Yes

3. Have the authors made all data underlying the findings in their manuscript fully available?

Reviewer #1: Yes

Reviewer #2: Yes

Reviewer #3: Yes

Reviewer #4: No

4. Is the manuscript presented in an intelligible fashion and written in standard English?

Reviewer #1: Yes

Reviewer #2: Yes

Reviewer #3: No

Reviewer #4: Yes

5. Review Comments to the Author

Reviewer #1: Compelling new pipeline for personalized music generation that delivers both quantitative and human-centric gains. It yields an average preference boost evidence of improved model efficiency and listener satisfaction. Coupled with intuitive explanations of its kernel methods, a thoughtful comparison to simple baselines and reproducibility framework, the work stands out for balancing technical rigor with open-science transparency.

###########################

1) Content and Deployment Improvements

The abstract could benefit  focusing on informative and concrete details (could highlight specific numeric results to anchor the contribution; e.g. “20% lower loss” or “+1.3 points average rating”). Abstracts usually include the key message prominently, thus, the authors might consider to explicitly mention at least one improvement (e.g. efficiency or user rating gain) to give readers an immediate sense of impact. Likewise, it may help to list the paper’s main contributions or findings clearly. Currently, some journals now encourage a structured abstract (with headings such as Purpose/Method/Findings/Originality) so that readers instantly see the novel elements. For example, it could detail three main-claims (for example: (1) “We propose a [X%] more efficient embedding strategy”; (2) “We achieve [Y%] lower loss in simulation”; (3) “In a user study (n=16), preference scores improved by [Z points]”) empowering clarity, but also anchoring novelty and scale for results up-front. Including a concrete example result and a mini “roadmap” of contributions will make the reader quickly appreciate the advance.

Similarly, conclusion might benefit if it explicitly acknowledges limitations (sample size) and outlines next steps to end on a positive-holding suggestion for future work like "pilot studied have been promising with encouraging results, yet limited to 16 listeners. A larger and more diverse can extend these findings” By citing the sample size and calling for follow-on work, the conclusion would both temper the claims and show awareness of scale. Writing guides recommend discussing limitations and then proposing concrete follow-ups. In practice this could read: “These promising user-driven improvements, while achieved on a small pilot, point toward future work on larger, more diverse participant pools.” A balanced framing highlighting both achievements and scope can align with standard conclusion guide lined strengths empowering credibility.

###########################

2) Readability and Technical Clarifications

Even as full of content, some sentences in the draft might seem overly long and dense. For example, the abstract’s one-sentence description of the listener task tries to cover background, method, and goal all at once. Best practices tend to split complex ideas into shorter sentences. A sentence of around 30 words can usually be broken into two simpler ones without loss of meaning, as some frameworks note long-winded sentences “lack clarity” and recommends aiming for roughly 12–15 words per sentence for academic writing. Concretely, one could rewrite the abstract as two sentences: first, “Listeners rate a series of songs produced by our music generator", and then “our pipeline then selects the next song that will most informatively improve the model, reducing the number of queries needed.” Shorter sentences let the reader absorb one idea at a time, vastly improving clarity.

Similarly, define specialized terms at first use. The manuscript uses “random rotational embedding” before it explains it, which can confuse readers. General writing standards recommend spelling out any uncommon abbreviation or technical term the first time it appears. For example: “we propose a random rotation embedding (RRE), where….” improving subsequent understanding clearly for the reader to not wait for the section for a priori clarification. As style note, one could write the term in full first and follow with the acronym to ensure understanding (if it’s not an acronym per se, the same logic applies to any novel phrase or concept).

Finally, the flow of the introduction can be improved by breaking up large paragraphs and using clear headings. Long paragraphs that mix high-level motivation and deep technical detail can overwhelm readers. It is often recommended to use subheadings or paragraph breaks to organize content. For instance, the intro could be partitioned into subsections such as “motivation,” “challenges,” and “approach.” Headings and subheadings “enhance the readability” of long texts by guiding and clearly segmenting topics. Even a blank line to start a new paragraph when shifting focus, besides section or subsection headers become relevant to help readers by chunking the content (e.g. first explain why personalized music generation is important, then detail the modeling approach, then outline contributions) rather than presenting it all as an undifferentiated block.

###########################

3) Methodological Context and Baselines

The Methods section may benefit from intuitive explanations of technical formulas, kernels and prior models. When introducing kernel, the text shows unintuitive equation. It might help to say in plain language what it does. If the kernel uses automatic relevance determination, you could note that each input dimension has its own lengthscale, effectively letting the model adjust how wiggly the function becomes in that direction. A larger lengthscale means the output hardly changes with spec. input, and a small lengthscale means it varies a lot.  Separating smoothness for each feature might allow some focus on relevant dimensions. A citation or fast tutorial could be used simply phrasing documentation upon very large data, covariance input-dependency giving non-expert readers an intuitive picture of what the math detail is elaborating.

Likewise, the paper currently cites classical strategies and additive models, but could mention several emerging approaches in high-dimensionality. Sparsity priors might model automatically ignoring irrelevant inputs and prior sparse axis-subspaces alineation could induce prior-shrinkage upon each kernel lengthscale. Citing this shows awareness methodic reach on to projection turning off unimportant parameters - shrinking their lengthscale - adapting sparsity to the data. Briefly mentioning related ideas upon feature sparsity could place the chosen kernel and embedding approach in further context for cutting-edge new research upon high-dimensionality.

Regarding evaluation baselines, it could be relevant to consider a simple control/null method for comparison. A random search sequence allows a standard baseline for optimization even if it performs poorly, as reporting helps quantify how much the proposed embedding actually improves over doing no dimensionality reduction. In practice one might cite common practices. Including such a null model might procure somewhat more rigor: if even a naïve search outperforms no embedding, it underscores the need for advanced methods.

###########################

4) Broadening Vision and Future Directions

The paper focuses on personalization but could connect more explicitly to the latest music generation models. For example, recent transformer or diffusion based audio models achieve high realism from text prompts, but such usually lack personalization loops. The authors could note this gap upon state-of-the-art text-to-music modeling productive and rich audio content from prompts, yet not based on user feedbacking framing novelty – linking preference feedback to generation – against big models that focus on generic/brandable quality.

In the discussion or future work, it would be valuable to mention diffusion models explicitly. Recent work on personalized audio generation starts to incorporate user adaptation into text-to-music diffusion systems, hence inviting to integrate our pipeline into diffusion-based music generators to create real-time personalized soundscapes. Citing is apt to investigate personalization of text-to-music diffusion models in a few-shot setting, signaling awareness of emerging paradigms and suggests that the proposed optimization could plug into cutting-edge audio pipelines.

Another growth idea is evaluating creativity and diversity. Mode collapse (generating many similar samples) is a known risk in generative models. The authors could guard against it by measuring diversity in the top-ranked songs. For example, computing an entropy or inter-sample distance metric over the high-rated outputs would reveal if the model is covering a broad range of styles or just one flavor. In general, evaluations of creative models often include mode collapse metrics to ensure outputs are varied. As one guide explains, such metrics might be to measure whether a generative model is producing a limited subset of outputs, as lower mode collapse scores indicate better diversity. Including an intention regarding diverse computability scores over selected songs (using spectral variance, a learned feature space or other) might ensure optimization is not collapsing to single strengthening the work’s advancements upon creative breadth.

Finally, the human study could be enriched with qualitative feedback if available. In addition to numeric ratings, gathering open-ended comments or interview data can reveal how users perceive and engage with the generative process. Mixed-methods research has shown that combining quantitative ratings with qualitative user impressions is especially insightful for creative AI tools because both can help demonstrate which features of the user experience are improved by AI, as well as offering insights into why. Thus, even a few exemplar quotes from listeners (or common themes from interviews) would provide depth – highlighting user creativity, satisfaction, or confusion that the numeric scores alone can’t show. Adding this perspective helps align with good practices and strengthens the human-centric claims of the paper.

###########################

5) Transparency, and Reproducibility

It is commendable that approvals and consents are documented. To further align with open science norms, the authors should include an explicit data availability statement pointing out - if feasible - code and data (song files, evaluation scripts, and survey results) are available in a public repository; data that support findings of this study are openly available, or similar. Adopting this format would clarify that others can access the materials, listing shared artifacts: preprocessing scripts, trained model checkpoints, and evaluation notebooks. This could enhance open-access followup reproducibility standards.

Similarly, adding details on compute and reproducibility would be valuable. Readers appreciate knowing the hardware and environment used. Ideas like “experiments were run on NVIDIA Tesla T4 GPU Intel Xeon"; "typical optimization iterations took x seconds (∼y minutes per user) on this setup”, or similar, provide practical context. Transparency about compute is always welcome, while reproducing state-of-the-art models often requires significant resources: documenting your setup helps others plan replication. Methods might specify random seeds and hyperparameters considering reproducibility guide recommends logging all parameters, including the random seeds.

Currently participant demographics as age and gender (or musical experience) are hard to find in the paper. A brief demographic summary grounds the human study. Something like “our 16 participants (8 female, 8 male) were aged 23–43 (mean=30) and had varied musical backgrounds (range: 0–10 years of formal training)" helps readers judge generalizability making methodology section more complete and adheres to transparency standards.

Adjustments like clear data-code sharing, compute details, seeds, and demographics could enhance the manuscript’s rigor and reproducibility and show attention to openness allowing the community to verify and build on the work, aligning with current best practices in AI research.

###########################

Overall, the work's results from rigorous performance on to user feedback lay a solid foundation and a larger, more diverse participant pool, integration with state-of-the-art diffusion-based audio generators, and diversity-and-creativity metrics could in future strengthen the highlights. With a data-and-code sharing plan, detailed logs and participant demographics, this work not only advances personalized music modeling, but also sets a high bar for transparent and reproducibility in AI research.

Reviewer #2: Thank you for your efforts in producing this manuscript on random rotational embedding Bayesian optimization (ROMBO). Evaluation for the algorithm’s performance, and a subsequent “real-world” experiment for satisfaction levels against controls, argue quite convincingly for the effectiveness of this novel application.

It is basically well-written, although please revise for some inconsistent and inaccurate use of singular/plural cases in the written langauge.

Reviewer #3: The paper has potential but currently requires substantial revision in terms of experimental justification, clarity of hypotheses, interpretation of results, and language quality. I encourage the authors to revise thoroughly and consider the points above to improve the manuscript.

Major Concerns

This paper addresses a complex and ambitious topic, and the authors have invested significant time and effort into its development. However, the current structure of the manuscript makes it difficult to follow, and substantial revisions are required before the work can be considered for publication.

Sample Size and Experimental Design: The sample size is a major concern. With only 16 participants, the statistical power is very limited—even when using Bayesian approaches. In a simple within-subject design, a minimum of 20–30 participants with 30–50 trials per participant is generally expected, especially when relying on limited prior information. As the authors describe the study as a novel setup involving multiple new factors, the rationale for the small sample size is unconvincing. To adequately model individual variability and stimulus effects—particularly when exploring emotional responses—one would typically require 40–60 participants and at least 50–100 trials per participant.

The manuscript mentions 22 samples for the Sobol analysis and 52 samples for optimization. However, with eight people in each gender, uneven groups, this seems insufficient for drawing robust conclusions.

Lack of Clear Hypotheses: The paper does not include formulated hypotheses or assumptions, either before or following the results section. This absence makes it difficult to evaluate the coherence and direction of the research.

Conclusion and Interpretation: The conclusion appears vague and underdeveloped. While the authors suggest that their model "works," there is little in-depth explanation as to why this is the case or how the research and control groups differ meaningfully. More interpretation and reflection on the findings are needed.

Language and Writing Quality: The manuscript requires substantial language editing. There are grammatical issues, unclear phrasing, and inconsistent use of tenses—sometimes even within the same sentence. For example:

“After collecting the volunteer’s consent, they are told the following pieces of information.”

Such inconsistencies hinder readability and should be addressed throughout the paper.

Minor Concerns

Abstract:

The number of participants (N) should be stated in the abstract.

The abstract should more clearly summarize the main results and key conclusions of the paper.

Introduction:

The authors should define what is meant by "music generation." Are they referring to AI models, rule-based systems, generative adversarial networks, or traditional composition and music production?

The rationale for involving amateur musicians in music generation systems should be better justified.

The paper seems to emphasize a system with primarily commercial applications, akin to those used by Spotify, YouTube, or Suno (mentioned by the authors). This raises questions about the motivation and scope of the study.

Please clarify the type of Likert scale used. Later in the manuscript, a 1–10 scale is mentioned, but this should be introduced earlier.

Rather than listing “contributions,” it would be more appropriate to describe these as the aims or objectives of the paper.

Methods – Experimental Design and Procedure:

The definition of “music generations” remains unclear at this stage in the manuscript.

There appears to be a large gender imbalance in the sample.

Rather than stating "for two weeks," it would be clearer to write, for example, "participants were recruited in February 2025."

Please provide the participants’ mean age and standard deviation.

Was musical preference, training, or aptitude assessed? Any such background variables should be reported or at least acknowledged as potential confounds.

The sentence “The consent allowed the abandonment of the experiment, and all data was analysed anonymously” should be revised for clarity and accuracy. A better formulation would be: “Participants could withdraw from the experiment at any time without consequences, and all data were analysed anonymously.” Note that this is standard practice in human subjects’ research and does not necessarily need to be emphasized unless participant withdrawal occurred (in which case, it should be reported).

The role and use of the control group are somewhat confusing. Please elaborate on its function and how the results differ between the experimental and control groups.

Can the generated musical stimuli be accessed (e.g., via sheet music or audio recordings)? Providing access via a platform such as Figshare would enhance transparency and reproducibility.

While allowing participants to control the volume is reasonable, the manuscript should specify the approximate decibel levels and clarify who judged what was “uncomfortably loud”—the researcher or the participant?

Discussion:

A discussion section is lacking. The authors should interpret their findings considering the hypotheses (if any), previous research, and potential limitations.

Conclusion:

The conclusion is insufficient. It should summarize the key findings, their implications, and suggestions for future research directions.

Figures:

The use of a single bar chart to represent survey responses is inappropriate and misleading. Each survey item should be represented by a separate chart. Additionally, consider using more suitable software than Excel for figure creation to improve visual clarity and professionalism.

The other data figures were nice and were made using a different tool.

Reviewer #4: As a musician with a keen (albeit new since 2022) interest in artificial intelligence, I found this manuscript fascinating despite not always understanding all of the mathematics. I would have liked to see the data and analysis on Github prior to acceptance, but I understand your rationale for not wanting to share everything right away.

I hope you can read the notes I've made on various pages.

As interesting as this work was, and as much as I learned just by reading it, I do wonder what the purpose is of "artificially generating music." In general, why try to get a machine (or an algorithm or a gpt) to make music? Why isn't human generation of music enough?

On page 5 I do ask a specific question related to musical complexity: If human music taste is subjective (and it definitely is) why should complexity matter?

I am surprised that you do not mention the "uncanny valley" phenomenon as it applies to artificially generated sound, given that on page 16, you state that most participants do not explicitly like the generated songs. Maybe a follow-up study could be to find out WHY they don't like the songs?

Overall, the writing is clear, flows well, and I found explanations of concepts and terminology to be easily understood.

6. PLOS authors have the option to publish the peer review history of their article (what does this mean?). If published, this will include your full peer review and any attached files.

Reviewer #1: No

Reviewer #2: No

Reviewer #3: **Yes: **Ulvhild Færøvik

Reviewer #4: No

---

## [Author Response · Author response to Decision Letter 1]

29 Sep 2025

We include a 'Response to Reviewers' file with individual responses to each reviewer. Other comments have been addressed as instructed.

---

## [Decision Letter · Decision Letter 1]

16 Oct 2025

Random rotational embedding Bayesian optimization for human-in-the-loop personalized music generation

PONE-D-25-27090R1

Dear Dr. Marcos,

We’re pleased to inform you that your manuscript has been judged scientifically suitable for publication and will be formally accepted for publication once it meets all outstanding technical requirements.

Kind regards,

Tara Rajendran, MBBS, Ph.D., MFA

Guest Editor

PLOS ONE

Additional Editor Comments (optional):

Reviewers' comments:

Reviewer's Responses to Questions

**Comments to the Author**

1. If the authors have adequately addressed your comments raised in a previous round of review and you feel that this manuscript is now acceptable for publication, you may indicate that here to bypass the “Comments to the Author” section, enter your conflict of interest statement in the “Confidential to Editor” section, and submit your "Accept" recommendation.

Reviewer #2: All comments have been addressed

Reviewer #3: All comments have been addressed

Reviewer #4: All comments have been addressed

2. Is the manuscript technically sound, and do the data support the conclusions?

Reviewer #2: Yes

Reviewer #3: Yes

Reviewer #4: Yes

3. Has the statistical analysis been performed appropriately and rigorously?

Reviewer #2: Yes

Reviewer #3: Yes

Reviewer #4: Yes

4. Have the authors made all data underlying the findings in their manuscript fully available?

Reviewer #2: Yes

Reviewer #3: Yes

Reviewer #4: Yes

5. Is the manuscript presented in an intelligible fashion and written in standard English?

Reviewer #2: Yes

Reviewer #3: Yes

Reviewer #4: Yes

6. Review Comments to the Author

Reviewer #2: Thank you for conscientously and thoroughly addressing the issues raised in review of your manuscript. The manuscript is much improved in overall coherence and in the details changed.

Reviewer #3: Thank you for the replies and clarification the paper looks much better now and I am happy with the changes.

Reviewer #4: I am satisfied that the authors addressed all of the comments/concerns I had during the first review.

7. PLOS authors have the option to publish the peer review history of their article (what does this mean?). If published, this will include your full peer review and any attached files.

Reviewer #2: No

Reviewer #3: **Yes: **Ulvhild Færøvik

Reviewer #4: No

---

## [Editor Report · Acceptance letter]

PONE-D-25-27090R1

PLOS ONE

Dear Dr. Marcos,

I'm pleased to inform you that your manuscript has been deemed suitable for publication in PLOS ONE. Congratulations! Your manuscript is now being handed over to our production team.

Kind regards,

on behalf of

Dr. Tara Rajendran

Guest Editor

PLOS ONE